# Using high-frequency solute synchronies to determine simple two-end-member mixing in catchments during storm events

**Nicolai Brekenfeld[1], Solenn Cotel[2], Mikaël Faucheux[1], Paul Floury[3], Colin Fourtet[2], Jérôme Gaillardet[4], Sophie Guillon[5], Yannick Hamon[1], Hocine Henine[6], Patrice Petitjean[7], Anne-Catherine Pierson-Wickmann[7], Marie-Claire Pierret[2], and Ophélie Fovet[1]**

[1]INRAE, Institut Agro, UMR SAS, Rennes, 35042, France
[2]ITES Institut Terre et Environnement de Strasbourg, CNRS/Université de Strasbourg, Strasbourg, 67000, France
[3]Extralab, Paris, 91400, France
[4]Institut de Physique du Globe de Paris, Université de Paris, CNRS, Paris, 75238, France
[5]Centre de Géosciences, MINES ParisTech, PSL University, Fontainebleau, 77300, France
[6]INRAE, University of Paris Saclay, UR HYCAR, Antony, 92761, France
[7]Géosciences Rennes UMR CNRS6118, University Rennes, Rennes, 35042, France

**Correspondence:** Nicolai Brekenfeld (nicolai.brekenfeld@gmail.com) and Ophélie Fovet (ophelie.fovet@inrae.fr)

**Abstract.** Stream water chemistry at catchment outlets is commonly used to infer flow paths of water through catchments and to quantify the relative contributions of various flow paths and/or end-members, especially during storm events. For this purpose, the number and nature of these flow paths or end-members are commonly determined with principal component analysis based on all available conservative solute data in inverse end-member mixing analyses (EMMAs). However, apart from the selection of conservative solutes, little attention is paid to the number and choice of the solutes that are included in the analysis, despite the impact this choice can have on the interpretation of the results from an inverse EMMA. Here, we propose a methodology that tries to fill this gap. For a given pair of measured solutes, the proposed methodology determines the minimum number of required end-members, based on the synchronous variation of the solutes during storm events. This allows identification of solute pairs for which a simple two-end-member mixing model is sufficient to explain their variation during storm events and of solute pairs, which show a more complex pattern requiring a higher-order end-member mixing model. We analyse the concentration–concentration relationships of several major ion pairs on the storm-event scale, using multi-year, high-frequency ($< 60\,\mathrm{min}$) monitoring data from the outlet of two small (0.8 to $5\,\mathrm{km}^2$) French catchments with contrasting land use, climate, and geology. A large number of storm events (56 % to 79 %) could be interpreted as being the result of a mixture of only two end-members, depending on the catchment and the ion pairs used. Even though some of these results could have been expected (e.g. a two-end-member model for the $Na^+/Cl^-$ pair in a catchment close to the Atlantic coast), others were more surprising and in contrast to previous studies. These findings might help to revise or improve perceptual catchment understanding of flow path or end-member contributions and of biogeochemical processes. In addition, this methodology can identify which solute pairs are governed by identical hydro-biogeochemical processes and which solutes are modified by more complex and diverse processes.

## 1 Introduction

Variations of stream water solute concentrations during storm events have been studied for several decades because of the associated large solute fluxes and their potential effect on aquatic organisms. In addition, high-frequency time series of stream water concentrations have frequently been used to characterize the event-scale hydrological and biogeochemical processes (sources, flow paths, and reactions) or the

ecosystem responses to nutrient inputs, which are considered useful for water resource management (Bieroza et al., 2023; Hill, 1993; Rode et al., 2016). For this characterization, water isotopes, dissolved ions, and other solutes serve as tracers of specific source areas or as indicators of residence times and chemical processes within the catchment. In recent decades and years, the use of auto-samplers and the development of in situ sensors, bank-side analysers, and in-the-field laboratories have allowed us to measure an ever-growing number of solutes at sub-daily to sub-hourly frequencies (Floury et al., 2017; Knapp et al., 2020; Rode et al., 2016). These high-frequency time series of the multi-elemental stream water chemistry reveal complex – and sometimes unpredictable – patterns of water sources and flow paths on intra- and inter-event scales (Knapp et al., 2020; Neal et al., 2012).

A variety of methods exist to interpret high-frequency time series of stream water chemistry. Probably the most frequently used methods are $c - Q$ (concentration–discharge) analysis and EMMA (end-member mixing analysis), depending on the scientific or operational question and on the additional available data. Without going into too much detail, event or seasonal $c - Q$ analysis is used to infer the relative source location of the solute (proximal vs. distal) and its transport or production mechanisms within the catchment (chemostatic vs. chemodynamic, with flushing and dilution) (Evans and Davies, 1998; Godsey et al., 2009). Using the $c - Q$ analysis, however, no direct conclusion can be drawn about the exact location of the solute sources. In contrast, the EMMA approach quantifies the contribution of identified water sources (end-members) based on the chemical signature of these sources and the assumed conservative behaviour of the solutes (Christophersen et al., 1990; Hooper et al., 1990).

The number of solutes used in EMMA studies is rarely of primary interest. In a forward-type EMMA (Christophersen and Hooper, 1992), the number and identity of the water sources (end-members), whose contributions to streamflow are to be quantified, are known or determined based on previous knowledge of the catchment. The solutes that differentiate between these sources are then selected in a second step (Durand and Juan Torres, 1996; Gillet et al., 2021; Ladouche et al., 2001). Alternatively, in the inverse-type EMMA (Christophersen and Hooper, 1992), the variation in stream chemistry is used as the primary information and is subsequently used to estimate the potential end-members and their numbers (Barthold et al., 2017; Christophersen and Hooper, 1992; Hooper, 2003; James and Roulet, 2006). Recently, further developments of the inverse-type EMMA, such as the convex-hull EMMA (CHEMMA), allowed even estimation of the chemical signatures of the potential end-members and not only their numbers (Xu Fei and Harman, 2022).

The forward-type EMMA requires a lot of prior knowledge about the catchment in question. It, therefore, cannot be applied easily to catchments that lie outside long-term observatories or to experimental catchments. In the inverse-type EMMA and the CHEMMA, this limitation is strongly relaxed. However, besides a selection of conservative solutes, no general, objective approach is used to select which solutes and how many should be included in the analysis of an inverse-type EMMA or CHEMMA. However, using a greater number of solutes also leads to a greater number of potential end-members that are needed to explain the observed variation of the stream chemistry (Barthold et al., 2011). Therefore, the selection of the solutes (and their number) used in an inverse EMMA or a CHEMMA likely has important implications for the interpretations and conclusions but is rarely discussed in detail, as mentioned by Lukens et al. (2022).

Here, we propose a new methodology to analyse high-frequency, multi-elemental time series of stream water that tries to overcome the constraints of the forward EMMA, the inverse EMMA, and the convex-hull EMMA, notably the prior selection of the conservative solutes. The proposed method relies on pairs of solutes in bivariate concentration–concentration plots (C–C plots) for storm events. These bivariate relationships can be used to identify synchrony between two solutes during storm events. When such synchrony is observed, a two-end-member system would be sufficient to explain the variability of the two solutes in the stream. In these cases, the potential end-members can be described as "diluted" or "unreacted" end-members (not to be confused with event water) on the one hand and "concentrated" or "reacted" end-members on the other hand (Lukens et al., 2022).

We define an end-member as a "water mass (e.g. riparian zone water, macro-pore solution, soil layer solution, groundwater, throughfall) with a distinct chemistry and with a distinct variation of its contribution". Furthermore, as in Hooper et al. (1990), we defined "end-members" as contributing sources that have extreme chemistry ("chemical boundaries"). Those end-members that can be formed by a mixture of two (or more) other end-members are not considered to be end-members. A definition of a "distinct variation of its contribution" is required, because two chemically distinct end-members that exhibit the same variation of their contribution appear in the stream only as one end-member with one chemical signature and not as two end-members. This is the case if the contribution of one of those end-members, $Q_{\mathrm{EM1},t}$, is a constant multiple of the contribution of the other end-member $Q_{\mathrm{EM2},t}$ (Eq. 1):

$$Q_{\mathrm{EM1},t} = Q_{\mathrm{EM2},t} \times k, \tag{1}$$

where $Q$ is the flow contribution ($\mathrm{L^3\,T^{-1}}$) of end-member 1 (EM1) or end-member 2 (EM2) at time instant $t$, and $k$ is a constant (during a given storm event). If, for example, one end-member always (before and during an event) contributes double the amount of water of the other end-member, then their individual contributions cannot be calculated because their chemistries appear as one, non-distinguishable mixture in the stream.

This methodology has been elaborated on based on original datasets of concentration time series that are both high-frequency (23 to 45 min) and multi-elemental as they include all major ions. The overall aim of this work is to propose a systematic methodology applicable to the analysis and interpretation of a large chemical dataset covering contrasting hydrological events from a multitude of different catchments.

## 2 Material, methods, and site description

### 2.1 Description of the study sites

High-frequency, multi-elemental analyses of stream water chemistry were conducted in two contrasting headwater catchments in France: the Kervidy-Naizin and Strengbach catchments (Fig. 1). They differ in topography (flat vs. steep), land use (agriculture vs. forest), climate (temperate oceanic vs. temperate oceanic mountainous), and geology (schist vs. granite). Both catchments are long-term observatories and are part of the French OZCAR Critical Zone Study Network (https://www.ozcar-ri.org, last access: 27 August 2024).

#### 2.1.1 Kervidy-Naizin

The Kervidy-Naizin catchment ($5\,km^2$, ORE AgrHyS Observatory) is located in Brittany, western France ($47.95°$ N, $2.8°$ W), with an elevation between 90 and 140 m a.s.l. (Fovet et al., 2018). The topography consists of gentle slopes ($< 5\,\%$). The bedrock is composed of low-permeability schists (Upper Proterozoic) overlain by fractured and fissured layers. The weathered zone is between 1 and 30 m deep and has a total porosity of 40 % to 50 %. The soils (silty loams, Cambisols), with a depth of 0.5 to 1.5 m, are generally well drained, except in the bottom lands close to the streams, where hydromorphic soils (Luvisols) are found. Land use is dominated by agriculture (90 % of the catchment area) associated with relatively high levels of nutrient inputs. The crop types are approximately 30 % maize, 30 % other cereals, and 30 % grasslands. The cropping systems and rotations are closely associated with the livestock type, with most farms practicing pig and/or dairy farming and a few farms having no animals (Fovet et al., 2018). The climate is temperate oceanic, with average annual rainfall of $840 \pm 220$ mm, Penman potential evapotranspiration of $700 \pm 60$ mm, runoff of $330 \pm 190$ mm, and air temperature of $11.2 \pm 0.6\,°C$ (1994 to 2017). The stream frequently dries up in summer for up to several months (Fovet et al., 2018).

#### 2.1.2 Strengbach

The Strengbach catchment ($0.8\,km^2$, OHGE Observatory) is located in the Vosges Mountains in north-eastern France ($48.12°$ N, $7.11°$ E), with an elevation between 880 and 1150 m a.s.l. (Pierret et al., 2018). The topography consists of steep slopes (20 % to 30 %). The bedrock is mainly composed of Hercynian calcium-poor granite, with various levels of hydrothermal alteration and some micro-granite and gneiss outcrops. The weathered zone has a thickness of 1 to 9 m and is overlain by brown acidic to ochreous, coarsely grained podzols (roughly 1 m thick). Land use is dominated by planted forests (90 % of the catchment area) consisting of 80 % spruce trees and 20 % beech trees (Pierret et al., 2018). The climate is temperate oceanic mountainous, with average annual precipitation of 1380 mm (varying between 900 and 1710 mm), Penman potential evapotranspiration of 570 mm (varying between 520 and 730 mm), runoff of 760 mm (varying between 490 and 1130 mm), and air temperature of $6\,°C$ (1986 to 2015) (Pierret et al., 2018; Strohmenger et al., 2022). Snowfall occurs in 2 to 4 months per year. The stream is fed by several intermittent and permanent springs, of which four permanent ones are used for drinking water supply by the nearby village. The long-term, annual runoff coefficient (runoff plus drinking water) is 0.55 to 0.60.

### 2.2 Data acquisition

High-frequency (every 25 to 45 min), multi-elemental (major cations and anions) analyses of the stream water chemistry at the outlets of the two catchments were conducted by automated stream-bank field laboratories. A detailed description can be found in Floury et al. (2017). The two field laboratories used at the two study sites were largely identical, differing slightly from the original system (Floury et al., 2017). Here we briefly describe their acquisition system for the time series of the major cations ($Na^+$, $Mg^{2+}$, $K^+$, $Ca^{2+}$) and anions ($Cl^-$, $NO_3^-$, $SO_4^{2-}$). It consisted of three main parts: (1) supply of unfiltered stream water to the field laboratory, (2) filtration, and (3) analysis of the filtered water by an ion chromatography system.

The field laboratories were located on the bank side at the outlet of the catchments. Stream water was continuously pumped by a surface pump to the field laboratory at a flow rate of around 400 to $600\,L\,h^{-1}$. The distances between the streams and the field laboratories were only a few metres ($< 10$ m) at Kervidy-Naizin but around 141 m at Strengbach. The mean transfer time of the water from the stream to the field laboratories was equal to or less than 8 min (determined with salt injections).

A small fraction of the stream water was continuously filtered with a two-step filtration system. The first filtration step consisted of a stainless-steel tangential filter ($0.5\,\mu mol\,L^{-1}$) with an automatic and regular cleaning mechanism (every few minutes) and a flow rate of roughly 0.5 to $10\,L\,h^{-1}$ (depending on the site). This was followed by the second filtration step, consisting of a mixed cellulose ester membrane filter ($0.22\,\mu mol\,L^{-1}$) that was manually replaced weekly (Kervidy-Naizin) or bi-weekly (Strengbach), in most cases, and with a flow rate of $0.1\,L\,h^{-1}$ or less. Due to the clogging of the membrane filter and the occasional extended periods

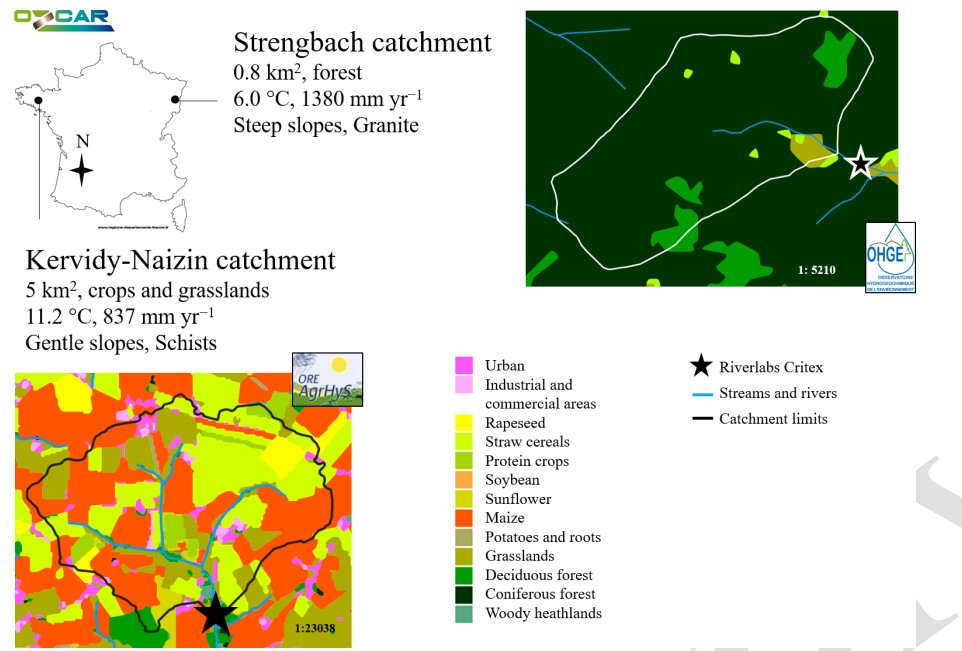

**Figure 1.** Study sites of Kervidy-Naizin (bottom left) and Strengbach (top right).

without a filter replacement (> 1 week at Kervidy-Naizin and > 2 weeks at Strengbach), transfer times of the filtered water to the analytical instrument reached up to 3 h.

The filtered water was analysed using a Dionex ICS-5000 Ion Chromatography System (Thermo Scientific) for the major cations and anions every 35 to 45 min at Kervidy-Naizin and every 20 to 30 min at Strengbach. The range of the time intervals between two consecutive analyses at each site was due to an optimization of the analytical procedure in the course of the project. The Ion Chromatography System was calibrated, on average, monthly at Naizin and every 3 months at Strengbach and after modifications to the instrument (replacement of consumables or capillaries, etc.). Validation with standards was conducted once a month on average. The limit of quantification was estimated to be around $1.0\,\mathrm{mg\,L^{-1}}$ (for the anions and cations) at Naizin and around $1.0\,\mathrm{mg\,L^{-1}}$ (for the anions) and $0.5\,\mathrm{mg\,L^{-1}}$ (for the cations) at Strengbach.

Stream discharge at both sites was estimated at the catchment outlets every minute (Kervidy-Naizin) and every 2 min (Strengbach) based on long-term, well-established rating curves and continuous water level monitoring.

Since both catchments are part of long-term observatories, we also used precipitation, throughfall, and groundwater data for the discussion of the results. Details of these measurements can be found in Fovet et al. (2018) and Pierret et al. (2018). The open-field precipitation chemistry (major anions and cations) has been measured monthly in bulk samples at Kervidy-Naizin since 2013 and fortnightly at Strengbach since 2005. In addition, throughfalls in a beech stand and a spruce stand have been measured fortnightly in bulk samples

at Strengbach since 1986. At Kervidy-Naizin, groundwater chemistry (major anions) has been measured in piezometers 3 to 8 m deep, at various locations, four times per year since 2000 (Fovet et al., 2018). The cation concentrations in the same piezometers were only measured irregularly as part of specific research projects between 2000 and 2008. The chemistry of the soil solutions in bottom-land areas has been measured irregularly with zero-tension lysimeters since 2011 during various projects at Kervidy-Naizin (Fovet et al., 2018).

### 2.3   Data treatment and analysis

The focus of this article is on presenting a methodology to analyse and interpret the variation of the stream chemistry during storm events. We therefore only present data of storm events. The following paragraphs outline the data acquisition period as well as the selection procedure of the storm events.

The field laboratories were operational from June 2018 onwards at Kervidy-Naizin and from November 2020 onwards at Strengbach. Here, we present the analysis of data collected until August 2022 in both catchments. Gaps in the dataset were due to various technical problems, i.e. failure of the main pump, power cuts due to lightning, chemical alteration of the filtered water that was likely due to the filtration system, problems with the analytical instrument, periods without technical support, and the COVID-19 pandemic. To ensure reliable data quality, only periods and storm events without technical issues were selected for further analysis.

Storm events were detected semi-automatically, based on the stream discharge time series. The exact parameters

and thresholds used in the procedure described below were adapted to each catchment. In the first step, the onset of a storm event was roughly detected by searching for periods of sustained, increasing streamflow, based on the first derivative of the flow. The precise starting date and time of the event, $T_0$, was subsequently determined by searching for the minimum streamflow within a few hours preceding the period of sustained, increasing flow. The peak of the event was then determined with the maximum streamflow within a few days after $T_0$ or before the onset of the next event, whichever occurred first. The end of the event was defined using a combination of the time after the peak of the event and the streamflow relative to the initial streamflow. This event-detection procedure was run automatically using an R script (RStudio Team, 2019). All detected events were then verified manually. Finally, only storm events not impacted by the regular maintenance of the field laboratory (cleaning, calibration, filter replacement) were selected.

At Kervidy-Naizin, we selected and analysed a total of 39 storm events with reliable and complete chemical data between June 2018 and April 2022 (out of 233 events detected based on discharge, resulting in a 17 % success rate for our field laboratory). At Strengbach, we selected and analysed 23 storm events between November 2020 and September 2022 (out of 212 events detected based on discharge: an 11 % success rate). At both sites, the selected storm events are unevenly distributed over the experimental period but cover a range of discharge values in different seasons and under different hydrological conditions. The timeline of the selected events can be seen in Fig. A1.

In order to compare the representativeness of the selected events, we compared the initial discharge (as a proxy for the season) and the maximum discharge increase during the event (as a proxy for the event magnitude) for the selected and detected events. At Kervidy-Naizin the median initial discharges were $68 \, \mathrm{L \, s^{-1}}$ (29, 39, 163, and $236 \, \mathrm{L \, s^{-1}}$ for the 10th, 25th, 75th, and 90th percentiles, respectively) and $62 \, \mathrm{L \, s^{-1}}$ (9.4, 31, 141, and $190 \, \mathrm{L \, s^{-1}}$ for the 10th, 25th, 75th, and 90th percentiles, respectively) for the selected and detected events, respectively. The median discharge increases were $60 \, \mathrm{L \, s^{-1}}$ (13, 25, 128, and $254 \, \mathrm{L \, s^{-1}}$ for the 10th, 25th, 75th, and 90th percentiles, respectively) and $36 \, \mathrm{L \, s^{-1}}$ (3.4, 7.2, 176, and $348 \, \mathrm{L \, s^{-1}}$ for the 10th, 25th, 75th, and 90th percentiles, respectively) for the selected and detected events, respectively. At Strengbach the median initial discharges were $4.6 \, \mathrm{L \, s^{-1}}$ (1.9, 2.6, 7.3, and $22 \, \mathrm{L \, s^{-1}}$ for the 10th, 25th, 75th, and 90th percentiles, respectively) and $7.4 \, \mathrm{L \, s^{-1}}$ (1.9, 2.7, 16, and $32 \, \mathrm{L \, s^{-1}}$ for the 10th, 25th, 75th, and 90th percentiles, respectively) for the selected and detected events, respectively. The median discharge increases were $5.2 \, \mathrm{L \, s^{-1}}$ (1.0, 2.1, 16, and $36 \, \mathrm{L \, s^{-1}}$ for the 10th, 25th, 75th, and 90th percentiles, respectively) and $3.1 \, \mathrm{L \, s^{-1}}$ (0.5, 1.2, 6.9, and $21.1 \, \mathrm{L \, s^{-1}}$ for the 10th, 25th, 75th, and 90th percentiles, respectively) for the selected and detected events, respectively.

## 2.4 Development of a methodology for a concentration–concentration typology

### 2.4.1 Event-scale solute behaviour classification based on C–C plots

We propose a methodology to analyse and interpret the variation of multi-elemental stream concentrations during storm events. This methodology focuses on identifying pairs of solutes that show synchronous variations during storm events. For each possible combination of two solutes and for each event, the concentration of the first solute as a function of that of the other solute (C–C plot) was examined (Fig. 2), the temporal dimension thus being lumped with the curve trajectory. We classified the observed C–C patterns into three types: synchronous, complex, and invariant. We defined a variation pattern as synchronous when the two solute concentrations have a linear relationship with a coefficient of determination ($R^2$) larger than or equal to 0.8 and when at least one of the solutes varies by at least 10 % relative to the maximum concentration. Complex variation patterns were based on a coefficient of determination smaller than 0.8. In addition, as for the synchronous pattern, at least one of the solutes had to vary by at least 10 %. Finally, invariant variation patterns were those where both of the solute concentrations varied by less than 10 % relative to the maximum concentration. The choice of the two used thresholds ($R^2 \geq 0.8$ and 10 % variation) was primarily based on the precision of our field laboratories. We will come back to this point in the discussion in Sect. 4.3.1. Examples of each type from the two catchments are shown in Fig. 2. Thus, for a given solute pair, all three variation types could be observed during different storm events.

For each storm event and solute pair, we used the function lm() in RStudio (RStudio Team, 2019) to calculate and the function summary() to extract the coefficient of determination between the variations of the two ion concentrations.

It should be noted here that the variable and unknown transfer time of the water sample from the stream to the analytical instrument was identical for all (seven) solutes analysed in this study ($Na^+$, $K^+$, $Ca^{2+}$, $Mg^{2+}$, $Cl^-$, $NO_3^-$, $SO_4^{2-}$) and was therefore not considered.

### 2.4.2 Rationale of the proposed methodology

The classification of the solutes into the three types described above provides the possibility of drawing conclusions about hydro-biogeochemical processes and the activation of contributing areas of the catchments during storm events. Synchronous solute pairs indicate that only two end-members are required to explain the variation of these solutes for a particular storm event. On a storm-event scale, this could be interpreted as the mixing of one pre-event end-member, which provides baseflow, with only one event-activated end-member.

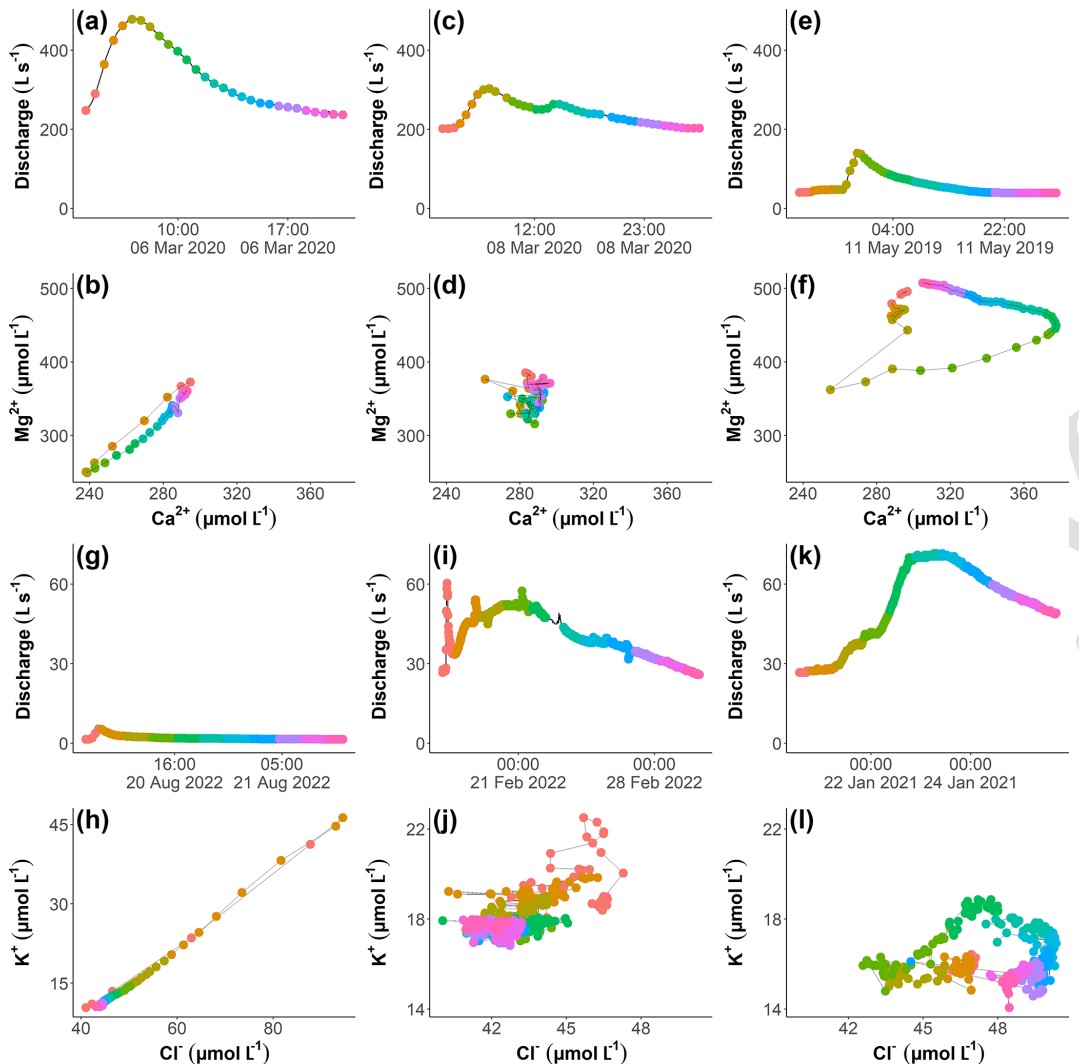

**Figure 2.** Hydrographs (**a**), (**c**), (**e**), (**g**), (**i**), and (**k**) and the corresponding concentration–concentration plots in panels (**b**), (**d**), and (**f**) (for $Mg^{2+}/Ca^{2+}$) and in panels (**h**), (**j**), and (**l**) (for $K^+/Cl^-$) for three events at Kervidy-Naizin (**a**–**f**) and Strengbach (**g**–**l**). Note in the fourth row that the scales of the axes in panel (**h**) are larger than those of the plots in panels (**j**) and (**l**). The concentration–concentration patterns are linear (**b, h**), invariant (**d, j**), and complex (**f, l**). Points with the same colour correspond to the same measurement time during the event for the hydrograph and the corresponding concentration–concentration graph, and they indicate the progression of the event (red: start; pink: end).

The interpretation of the solutes in the other two types (invariant and complex variation types) is more ambiguous. Invariant solutes can be interpreted either (1) as the existence of two or more end-members which have identical chemical signatures or (2) as a flow path reactivity that modifies and matches the concentrations of initially different end-members on their flow path towards the stream. This second process was observed with a sprinkling experiment (Anderson et al., 1997) and a leaching experiment (Hill, 1993), although in catchment contexts that might not be completely comparable with our field sites. The former experiment was conducted in a very steep (40–45°), 860 m² small, and forested non-channelled valley (Anderson et al., 1997),

whereas the latter one used samples from the surface peat and needleleaf litter from a groundwater-connected headwater swamp (Hill, 1993). Complex solutes can be caused by (1) mixing of more than two end-members or (2) variations of the end-member concentrations in the course of a storm event. The former includes the case in which different end-members mix along the flow path towards the catchment outlet if these end-members differ in their chemistry and vary in their relative contribution to streamflow (see further details of the definition of an end-member in the "Introduction" section).

### 2.4.3 Evaluation of the variation of the molar ratios on the event scale

For the synchronous solute pairs, we analysed whether and to what extent the ratio of the two solutes of the different solute pairs changed in the course of individual storm events. This indicates whether and to what extent the potential end-members also have different solute ratios. Analysis of the variation of the solute ratios in the stream and the potential end-members can be used to draw conclusions about some hydrological and (bio-)geochemical processes in the catchment. In general, processes that act on both solutes to the same degree do not change the solute ratio. This is the case, for example, for evaporation (assuming that the majority of the solute mass is not evaporated), transpiration (for solutes that are not taken up by the plants), and dilution by diluted rainwater (for solutes that are found at very low concentrations in the rainwater). In contrast, processes that act preferentially on one solute compared to the other one will result in a changing solute ratio. This is the case, for example, for external inputs (e.g. fertilizer applications in agricultural catchments), preferential uptake or release by organisms (e.g. nutrients), weathering (release of solutes from primary or secondary minerals), and precipitation (formation of secondary minerals).

To evaluate the evolution of the solute ratios during the storm events, we calculated the ratio at the beginning of the storm event (the "initial ratio") and the ratio at the time of the maximum concentration increase or decrease (the "peak ratio"). The initial ratio was calculated by taking the arithmetic mean of the first two measurements during a storm event. The peak ratio was calculated as the ratio at the time when the denominating ion (e.g. $Na^+$ of the $Cl^-/Na^+$ ratio) was at its extreme value, i.e. at its minimum concentration or at its maximum concentration. This analysis was only applied to synchronous ions. Therefore, both the ion in the numerator and the ion in the denominator could be used to calculate the peak ratio, leading to the same results. Our choice to use the ion in the denominator was arbitrary.

### 2.4.4 Evaluation of the inter-event variability

In addition to the solute variations on the event scale (concentrations and ratios), we evaluated the variations of the solute pairs across all the different events (i.e. inter-event variability). This analysis can reveal whether the chemistry of the end-members varies across the different events (i.e. across different hydrological states and seasons).

The inter-event synchrony of a solute pair compares the concentration decreases and increases of one solute across all analysed storm events with the inter-event concentration decreases and increases of another solute. This method allows evaluation, for example, of whether, during a particular storm event, a minor concentration decrease of one solute also leads to a minor concentration decrease of another

solute and whether this correlation is observed across different storm events. For this purpose, we used the relative concentration variations. For each storm event $i$ and each ion $j$, we calculated minimum $((C_{i,j,\text{min}}-C_{i,j,\text{start}})/C_{i,j,\text{start}})$ and maximum $((C_{i,j,\text{max}}-C_{i,j,\text{start}})/C_{i,j,\text{start}})$ concentration changes relative to the initial concentrations at the start of the storm events.

## 3 Results

### 3.1 General chemical variation during the selected storm events

The chemical variations during the 39 storm events at Kervidy-Naizin and the 23 storm events at Strengbach are ion- and catchment-specific and depend on the season and the hydrological conditions. Three examples are presented for each catchment in Fig. B1. On the event scale, individual ion concentrations showed contrasting evolutions: they (1) remained constant (e.g. $Na^+$, $Mg^{2+}$, and $Ca^{2+}$ at Strengbach), (2) primarily decreased (e.g. $Na^+$, $Cl^-$, $Mg^{2+}$, and $NO_3^-$ at Kervidy-Naizin), (3) primarily increased (e.g. $K^+$ and $Cl^-$ at Strengbach), or (4) increased and decreased during the same event (e.g. $Ca^{2+}$ and $SO_4^{2-}$ at times at Kervidy-Naizin). These ion-specific patterns are either consistent between events or differed from event to event, as illustrated in Fig. B1.

### 3.2 Event-scale concentration–concentration pattern

In this section and the following ones, we group the ion pairs into one of the three types of concentration–concentration patterns (synchronous, complex, or invariant) based on their dominant (absolute or relative) inter-seasonal patterns during the storm events. Due to this generalization, individual storm events can deviate from the overall pattern of a given solute pair.

At Kervidy-Naizin, the six solute pairs ($Cl^-/Na^+$, $Cl^-/Mg^{2+}$, $Cl^-/NO_3^-$, $Na^+/Mg^{2+}$, $Na^+/NO_3^-$, and $Mg^{2+}/NO_3^-$) that can be formed by the four solutes $Cl^-$, $Na^+$, $Mg^{2+}$, and $NO_3^-$ exhibited a synchronous variation pattern in over 56 % (using $R^2 > 0.8$) of the storm events (Table 1). This percentage was particularly high (at least 77 % of the events) for $Cl^-/Na^+$ and $Mg^{2+}/NO_3^-$ and lower (56 % to 64 %) for the four remaining solute pairs. For Kervidy-Naizin, we will therefore refer to these four solutes as synchronous.

For $SO_4^{2-}/Ca^{2+}$ at Kervidy-Naizin, a synchronous variation was observed during 33 % of the storm events. The solute pairs combining $SO_4^{2-}$ with the synchronous ions ($Na^+$, $Cl^-$, $Mg^{2+}$, and $NO_3^-$) exhibited synchronous variation in 21 % to 31 % of the storm events (Tables 1 and C1). This percentage was similar (26 % to 28 %) for the pairs made of $Ca^{2+}$ with synchronous ions. Potassium exhibited a synchronous variation with the other solutes during no more than

**Table 1.** Number (and percentage) of storm events with synchronous, complex, and invariant concentration–concentration patterns for different pairs of solutes for Strengbach and Kervidy-Naizin. For the synchronous concentration–concentration pattern, the values are listed for two thresholds ($R^2 \geqq 0.8$ and $R^2 \geqq 0.9$). Listed are all solute pairs that exhibit a synchronous variation ($R^2 > 0.8$) in at least 30 % of the storm events. All other solute pairs are listed in Tables C1 and C2.

| Ion pairs | Strengbach ($n = 23$) | | | | Kervidy-Naizin ($n = 39$) | | | |
|---|---|---|---|---|---|---|---|---|
| | Synchronous | | Complex | Invariant | Synchronous | | Complex | Invariant |
| | $R^2 \geqq 0.9$ | $R^2 \geqq 0.8$ | | | $R^2 \geqq 0.9$ | $R^2 \geqq 0.8$ | | |
| $K^+/Cl^-$ | 10 (43 %) | 13 (57 %) | 4 (17 %) | 6 (26 %) | | | | |
| $Ca^{2+}/Mg^{2+}$ | 4 (17 %) | 8 (35 %) | 7 (30 %) | 8 (35 %) | | | | |
| $NO_3^-/Mg^{2+}$ | | | | | 27 (69 %) | 31 (79 %) | 8 (21 %) | 0 (0 %) |
| $Cl^-/Na^+$ | | | | | 28 (72 %) | 30 (77 %) | 0 (0 %) | 9 (23 %) |
| $Mg^{2+}/Na^+$ | | | | | 21 (54 %) | 25 (64 %) | 6 (15 %) | 8 (21 %) |
| $NO_3^-/Cl^-$ | | | | | 20 (51 %) | 24 (62 %) | 10 (26 %) | 5 (13 %) |
| $Mg^{2+}/Cl^-$ | | | | | 22 (56 %) | 23 (59 %) | 7 (18 %) | 9 (23 %) |
| $NO_3^-/Na^+$ | | | | | 20 (51 %) | 22 (56 %) | 13 (33 %) | 4 (10 %) |
| $SO_4^{2-}/Ca^{2+}$ | | | | | 10 (26 %) | 13 (33 %) | 7 (18 %) | 19 (49 %) |
| $SO_4^{2-}/NO_3^-$ | | | | | 7 (18 %) | 12 (31 %) | 21 (54 %) | 6 (15 %) |

3 % of the storm events, except for $Mg^{2+}$ (15 %) and $NO_3^-$ (18 %). Due to these low percentages, we do not consider $SO_4^{2-}$, $Ca^{2+}$, and $K^+$ to be synchronous solutes at Kervidy-Naizin. In contrast, $Ca^{2+}$ primarily showed a complex variation pattern (except for $K^+/Ca^{2+}$), $SO_4^{2-}$ primarily showed an invariant variation pattern (except for $SO_4^{2-}/NO_3^-$), and $K^+$ showed either complex or invariant variation patterns (Tables 1 and C1).

At Strengbach, synchronous variation during storm events was observed only for $K^+/Cl^-$ (57 %, Table 1). All other possible solute pairs exhibited synchronous variation during less than 20 % of the storm events, except for $Ca^{2+}/Mg^{2+}$ (35 %) and $Mg^{2+}/Na^+$ (26 %) (Table C2). The majority of the ion pairs, which included $Mg^{2+}$, $Na^+$, or $Cl^-$, exhibited a complex variation pattern (Table C2). In contrast, the other four solutes ($Ca^{2+}$, $SO_4^{2-}$, $NO_3^-$, and $K^+$) could not be attributed to one group, because they showed either complex or invariant variation patterns (Table C2).

### 3.3 Relative concentration variations on the event scale of all the solutes

The general event-scale variation of the different solutes is ion- and catchment-specific. At Kervidy-Naizin, consistent concentration decreases were observed on the event scale for the four synchronous ions ($Na^+$, $Cl^-$, $Mg^{2+}$, and $NO_3^-$) for almost all the events, with median concentration decreases of $-9$ % ($Na^+$) to $22$ % ($NO_3^-$) (Fig. 3). Consistent, event-scale mobilization patterns were observed for $Cl^-$ (median: $+45$ %) and $K^+$ ($+68$ %) at Strengbach as well as for $K^+$ ($+20$ %) at Kervidy-Naizin (Fig. 3). Inconsistent event-scale mobilization patterns were observed for $SO_4^{2-}$ ($-11$ %$/+10$ %) and to some degree for $Ca^{2+}$ ($-11$ %$/+3$ %) at Kervidy-Naizin and $NO_3^-$ ($-16$ %$/+13$ %) at Strengbach

(Fig. 3). These solutes exhibited increasing and decreasing concentrations during the same event (see an example in Fig. B1) or different patterns during different events.

For the other four solutes at Strengbach, median relative concentration variations remained small. Variations of less than 10 % were observed for $SO_4^{2-}$ ($-8$ %$/+2$ %; median decrease or median increase), $Na^+$ ($-6$ %$/+5$ %), $Mg^{2+}$ ($-5$ %$/+8$ %), and $Ca^{2+}$ ($-4$ %$/+6$ %).

### 3.4 Event-scale variation of molar ratios

For the synchronous solute pairs, we evaluated the variation of their ratios during individual storm events. In general, the molar ratio during the events evolved towards the median ratio in the rain (at Kervidy-Naizin) or in the throughfall (at Strengbach), indicating that the ratio of the rain was closer to the ratio of the event-activated water than to the ratio of the pre-event water (e.g. baseflow) (Fig. 4). It must be highlighted, however, that we are comparing the high-frequency ($< 1$ h) variability in the stream with the low-frequency bulk and long-term measurements in the rain (monthly at Kervidy-Naizin and fortnightly at Strengbach) and throughfall (biweekly at Strengbach).

At Kervidy-Naizin, the relative difference between the initial and peak ratios was small for $Cl^-/Na^+$ (median: $-3$ %) and only slightly higher for $Mg^{2+}/Cl^-$ and $Mg^{2+}/Na^+$ (medians: $-4$ % and $-7$ %, respectively). In contrast, for the three pairs that include $NO_3^-$, the median difference between the initial and peak ratios was larger (the medians range between $-12$ % and $-15$ %) (Fig. 4). This indicates that, during the storm events at Kervidy-Naizin, the relative concentration decrease was strongest for $NO_3^-$, followed by $Mg^{2+}$, $Na^+$, and $Cl^-$.

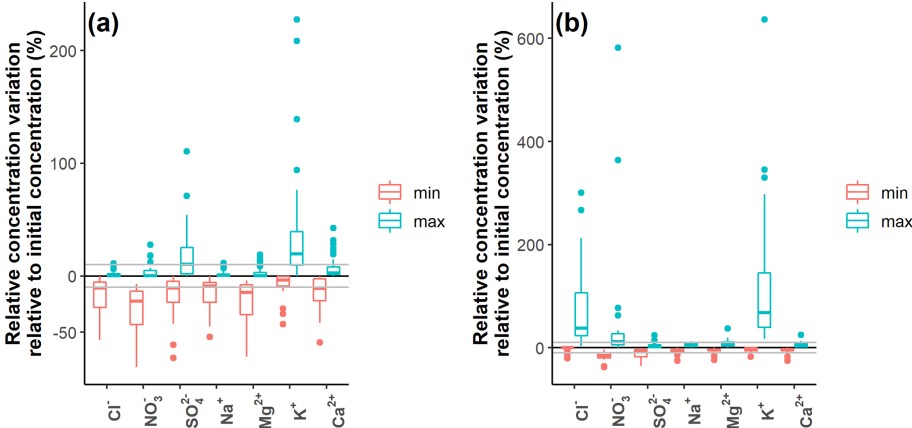

**Figure 3.** Relative concentration variations (percentage of the initial concentration) for Kervidy-Naizin **(a)** ($n = 39$ storm events) and Strengbach **(b)** ($n = 23$ storm events). For each storm event $i$ and each solute $j$, relative minimum (red) and maximum (green) concentration variations $C$ were calculated as $(C_{i,j,\text{min}} - C_{i,j,\text{start}})/C_{i,j,\text{start}}$ and $(C_{i,j,\text{max}} - C_{i,j,\text{start}})/C_{i,j,\text{start}}$, respectively. The grey horizontal lines indicate the 10 % thresholds. The patterns of solutes such as $SO_4^{2-}$ and $Ca^{2+}$ at Kervidy-Naizin (both with $> 10\%$ increases and decreases) are due to some events exhibiting a primarily decreasing concentration variation and others a primarily increasing concentration variation or are due to some events exhibiting both increasing and decreasing concentration variations during the same event.

At Strengbach, the ratio of $K^+/Cl^-$ increased (median: 15 %) during the events, getting closer to the ratio in the throughfall (Fig. 4).

### 3.5 Inter-event synchrony of concentration variations

The solute pairs that were synchronous on the event scale ($Na^+$, $Cl^-$, $Mg^{2+}$, and $NO_3^-$ at Kervidy-Naizin; $K^+/Cl^-$ at Strengbach) were also synchronous on the inter-event scale. Some examples are shown in Fig. 5; for the rest, see Figs. D1 and D2.

At Kervidy-Naizin, the inter-event correlation coefficients for all six pairs of the four synchronous solutes were at least 0.93, indicating a strong inter-event synchrony. $Ca^{2+}$ and $SO_4^{2-}$ (solutes with increasing and decreasing concentrations on the event scale) showed synchrony with the other synchronous ions for those events (or parts of them), which were *diluting* (corresponding to the turquoise points and values in Fig. D1). The correlation coefficients of $Ca^{2+}$ with the four synchronous ions ranged between 0.88 and 0.93 (Fig. D1), and those of $SO_4^{2-}$ ranged between 0.65 and 0.76 (Fig. D1).

At Strengbach, the inter-event correlation coefficient of the synchronous solute pair ($K^+/Cl^-$) was 0.93 (Figs. 5 and D2), indicating high synchrony on the inter-event scale. The inter-event correlation coefficients of $Mg^{2+}$, $Ca^{2+}$, $Na^+$, and $SO_4^{2-}$ ranged between 0.82 and 0.88, except for $Mg^{2+}/Ca^{2+}$ (0.96) (Fig. D2), equally indicating some degree of inter-event synchrony.

## 4 Discussion

### 4.1 Synchronous solute variation on the event scale

The synchronous variation of two solutes on the event scale can be interpreted as a mixture between only two end-members: one "reacted" or "concentrated" end-member and one "unreacted" or "diluted" end-member (Lukens et al., 2022). The aim is not to identify the different end-members but rather to conclude that only two different end-members are contributing to streamflow. However, at a later stage, these theoretical end-members could be mapped to some water masses.

These two end-members can be interpreted as one pre-event end-member which provides streamflow before and during the storm event and one event-activated end-member. The fact that only one event-activated end-member exists for a given solute pair indicates that the different contributing parts of the catchment that are activated during the storm event all have an equivalent chemical signature and are spatially homogeneous without chemical "hotspots". Hotspots, such as riparian zones with high reactivity, can be neglected, because they would otherwise likely constitute a second event-activated end-member. Therefore, the processes are also spatially homogeneous, which lead to the "reacted" or "concentrated" and "unreacted" or "diluted" end-members. These processes could be wet and dry deposition, evapotranspiration, weathering of primary minerals, formation and dissolution of secondary minerals, anthropogenic inputs, and biogeochemical transformations, to name only a few.

To interpret the synchronous solutes and the variation of their ratios, hydrological and (bio-)geochemical processes can be compared that change the solute ratio (e.g. input,

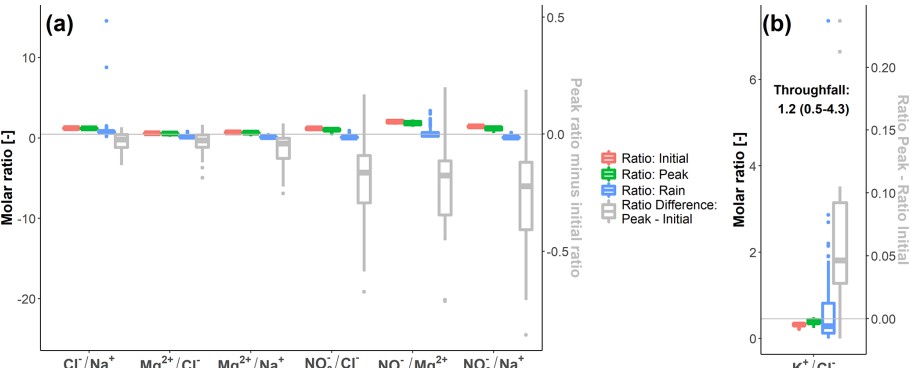

**Figure 4.** Concentration ratios at the beginning of the events and at the moment of the extreme value of the concentration decrease or increase ("Peak"), their pairwise difference (grey), and the ratios in the rain (blue) for Kervidy-Naizin **(a)** and Strengbach **(b)**. For Strengbach, the median and interquartile range are additionally given for the throughfall. Ratios are only shown for the synchronous solute pairs and for those events for which synchronous variations (i.e. $R^2 > 0.8$) were observed for the specific ion pair.

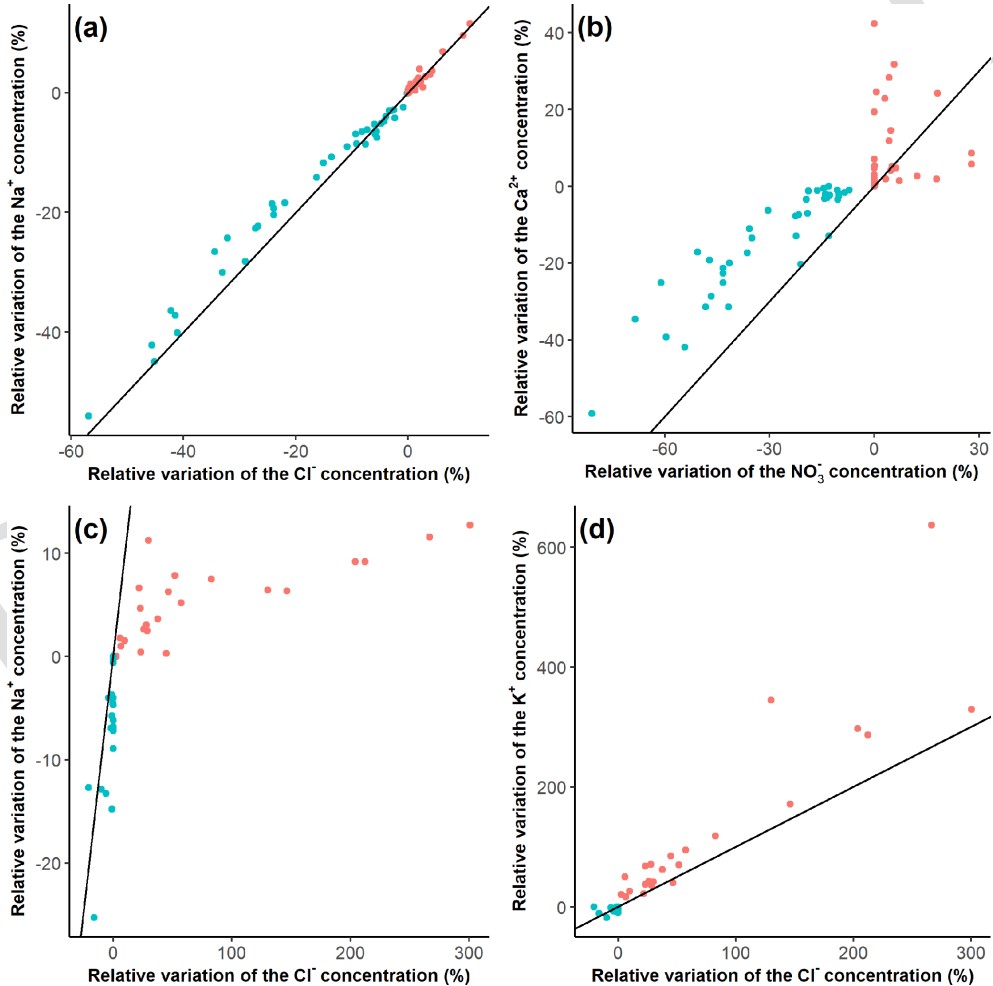

**Figure 5.** Inter-event synchrony for $Na^+/Cl^-$ and $Ca^{2+}/NO_3^-$ at Kervidy-Naizin (panels **a** and **b**, respectively) and for $Na^+/Cl^-$ and $K^+/Cl^-$ at Strengbach (panels **c** and **d**, respectively) as examples. All remaining pairs are shown in Figs. D1 and D2. Relative concentrations increase [red: $(C_{\mathrm{maximum},j} - C_{\mathrm{initial},j})/C_{\mathrm{initial},j}$] and decrease [turquoise: $(C_{\mathrm{initial},j} - C_{\mathrm{minimum},j})/C_{\mathrm{initial},j}$] for each event $j$ (%) in relation to the initial concentration. Therefore, there are two points for each event, one indicating the minimum (turquoise) relative concentration and one indicating the maximum (red) relative concentration. The black line is the 1 : 1 line.

production, or consumption of one of the solutes by fertilizer applications, weathering, or precipitation of secondary minerals, respectively) and those that keep it constant (e.g. evapotranspiration). These processes are, of course, solute-specific. For a solute pair with an observed solute ratio during storm events that remains constant, catchment processes that keep the ratio constant can be considered to be of primary importance, whereas those that change the ratio are likely negligible and vice versa. Therefore, this analysis provides a tool to evaluate for each solute pair which hydrological and (bio-)geochemical processes are likely the most important and which can be neglected. Finally, the information provided by the synchronous behaviour of a solute pair (only two end-members, non-importance of hotspots) and by the variation of the solute ratio (negligible and dominant processes) can change and improve the hydrological and (bio-)geochemical understanding of the catchment processes.

### 4.1.1 Agricultural catchment

At Kervidy-Naizin, the $Na^+/Cl^-$ ratio measured in the stream, in piezometers, or in soil solutions is very similar to and overlaps with the ratio in the rain. We interpret evapotranspiration as being the primary process leading to the elevated solute concentration in the "concentrated" end-member. This is due to the constant ratio observed across different catchment compartments and the fact that evapotranspiration is the main process that increases the concentrations of the solutes (more or less strongly during different seasons) but keeps their ratios constant (especially for solutes that are considered to not be recycled by the vegetation). This interpretation is in line with some previous interpretations from the same catchment (Ayraud et al., 2008) and inputs from anthropogenic activity ($Cl^-$) or weathering ($Na^+$) being negligible compared to input from precipitation. The apparently negligible anthropogenic input of $Cl^-$ is in contrast to interpretations of previous studies from the same catchment, where inputs of mineral KCl fertilizers by farmers were hypothesized (Aubert et al., 2013). This is similar for $Na^+$, for which input from weathering and soil leaching to the stream export was expected at Kervidy-Naizin. In a similar, granitic, and agricultural catchment in Brittany, a weathering input of over 50 % was estimated for $Na^+$ (Pierson-Wickmann et al., 2009), and increased $Na^+$ concentrations and $Na^+/Cl^-$ ratios were observed across different rivers in Brittany over the last decades, likely due to $NH_4$-oxidation-induced acidification and soil leaching (Aquilina et al., 2012). Based on these studies, the observed negligible inputs of $Cl^-$ and $Na^+$ from agricultural inputs and weathering, respectively, relative to the large inputs by precipitation, were unexpected.

At Kervidy-Naizin, the stream water ratios of $Mg^{2+}$ and $NO_3^-$ with the other synchronous ions are elevated compared to the rain ratios. The $Mg^{2+}/Cl^-$ and $Mg^{2+}/Na^+$ ratios in the stream and piezometers are around 5 times higher than in the rain, whereas the ratios of those solutes ($Mg^{2+}$, $Cl^-$,

$Na^+$) with $NO_3^-$ are even higher (10 times higher in the case of $Na^+$ and $Cl^-$ and 2 times higher in the case of $Mg^{2+}$). This indicates that there must be additional inputs of $Mg^{2+}$ and $NO_3^-$ due to weathering, soil leaching, liming, and/or (chemical) fertilizer applications relative to $Na^+$ and $Cl^-$. This was expected, especially for $NO_3^-$, due to the strong agricultural activity in the catchment. For $Mg^{2+}$, input due to soil leaching and weathering could have been expected as well, based on previous observations in Brittany (Aquilina et al., 2012; Pierson-Wickmann et al., 2009). From the observed synchrony between these solutes, which indicates a two-end-member system, we can conclude (1) that the inputs must be spatially homogeneous in order to lead to a two-end-member system and (2) that there are no hotspots within the event-activated parts of the catchment, such as wetlands stimulating denitrification, which would retain or mobilize $Mg^{2+}$ or $NO_3^-$ differently. For $NO_3^-$, for example, this indicates that the impact of potential hotspots of denitrification is likely negligible for the stream chemistry during storm events.

The observation that the ratios of ion pairs evolve towards lower ratios in rain during storm events indicates either partial dilution by rainwater or activation of an end-member, where the biogeochemical reactions have not proceeded as far as in reacted or concentrated end-members. In the latter case, this means that the concentrations of $NO_3^-$ and $Mg^{2+}$ relative to $Na^+$ and $Cl^-$ are lower than in the concentrated end-member. This could be expected for $Mg^{2+}$, for which significant input by rock weathering is assumed. For $NO_3^-$, with a larger agricultural input, this might seem to be less evident. However, it could indicate the importance of the large legacy effect of $NO_3^-$ in the deep groundwater, leading to a relatively higher concentration in the concentrated or reacted end-member (Molénat et al., 2002) or a relatively reduced concentration in the unreacted or event-activated end-member due to denitrification.

### 4.1.2 Forested catchment

At Strengbach, $K^+/Cl^-$ is the only synchronous solute pair, which can be explained by a preserved signal from throughfall. Particularly high concentrations and fluxes of $K^+$ in throughfall (averages of 31.6 and 22.4 $kg\,ha^{-1}\,yr^{-1}$ under beech and spruce plots, respectively) relative to rain (2.2 $kg\,ha^{-1}\,yr^{-1}$) and the export at the catchment outlet (5.5 $kg\,ha^{-1}\,yr^{-1}$) were observed in this catchment and were interpreted as an input from biological excretion of leaves (Pierret et al., 2019). In addition, K is the only chemical element whose fluxes and concentrations at the outlet at Strengbach are lower than its inputs (atmospheric plus throughfall), indicating the minor importance of the weathering fluxes and strong biogeochemical cycling in the forest. Similarly, but in smaller proportions, higher inputs of $Cl^-$ were also observed under beech (13 $kg\,ha^{-1}\,yr^{-1}$) and spruce trees (19 $kg\,ha^{-1}\,yr^{-1}$) in comparison with rain (6.4 $kg\,ha^{-1}\,yr^{-1}$), due to the dry interception by leaves and

needles (Pierret et al., 2019). As no identified minerals from the soils and granite contain $Cl^-$, the weathering fluxes of $Cl^-$ are considered negligible at Strengbach. We therefore hypothesize that the synchronous $K^+$ and $Cl^-$ peaks during storm events might be due to higher contributions of superficial fluxes strongly influenced by throughfall. Due to the large flux from throughfall, its signal might be observable in the stream. However, on its path to the stream, some of the $K^+$ ions are probably retained by vegetation or mineral surfaces, because the $K^+/Cl^-$ ratio in the stream and the tributaries is close to the ratio in the rain and lower than in the throughfall. This is underpinned by a study in the catchment that indicates that soil solutions show a reduction in the $K^+$ concentration with depth (a factor of almost 5 at depths between 5 and 60 cm) (Beaulieu et al., 2020), highlighting the strong recycling by the vegetation.

Similar observations (increasing $K^+$ and $Cl^-$ concentrations during storm events) were also reported from a small (4 ha), steep, forested catchment on low-grade metamorphic schist with a mountainous Mediterranean climate in northern Spain (Ávila et al., 1992). The authors attributed the low $K^+$ concentrations during baseflow to uptake by the vegetation and fixation into clay lattices and the increased storm event concentrations to mobilization from the canopy and the organic soil layer (Ávila et al., 1992), which might be applicable here too. However, despite the fact that the authors did not use the concentration–concentration behaviour of $K^+$ and $Cl^-$ to analyse their synchrony, as we propose here, the behaviour of $K^+$ and $Cl^-$ seems to be visually less synchronous than what we observed at Strengbach. Increasing, or variable, $K^+$ concentration behaviour during storm events was also reported from other steep, forested catchments and was attributed to mobilization from live or dead biomass or cation exchange buffering in the soil layer (Barthold et al., 2017; Knapp et al., 2020). However, to our knowledge, the largely synchronous behaviour of $K^+$ and $Cl^-$, which we observed at Strengbach and which indicates a two-end-member system, was not reported previously and was unexpected for us.

## 4.2 Inter-event synchrony

The inter-event synchrony evaluates whether different solute pairs show similar behaviours and intensities of their concentration decreases and/or increases across different storm events. For example, at Kervidy-Naizin, $Cl^-$ and $Na^+$ are synchronous on the inter-event scale: during small storm events, for example, when $Cl^-$ is diluted by 20 %, $Na^+$ is also diluted by 20 %, and during large storm events both are diluted by up to 60 %. This 1 : 1 dilution behaviour for $Cl^-$ and $Na^+$ agrees with the observation that their ratio remains almost constant during individual events. Other solute pairs, such as $Cl^-$ and $NO_3^-$, are synchronous on the inter-event scale, but they do not lie on the 1 : 1 line, because their ratio varies during individual storm events.

The observed inter-event synchrony of the synchronous ions in both catchments indicates that similar processes govern these solutes not only during individual storm events, but also across different events. Consistent variation patterns across different storm events for specific groups of ions were observed previously (Ávila et al., 1992). Storm events showing a strong concentration increase (at Strengbach) or decrease (at Kervidy-Naizin) of the synchronous ions can then either be interpreted as a large contribution of the event-activated end-member or as a chemistry of the event-activated end-member that changed relative to previous storm events. Changes in end-member chemistry linked to antecedent hydrological conditions, seasonal climatic variations, or microbial activities (Knapp et al., 2020) and variable end-member contributions across different storm events (Pierret et al., 2014) were also observed before. Overall, synchronous ions on the event scale and on the inter-event scale indicate that the hydro-biogeochemical processes that lead to the diluted and concentrated end-members are common for these ions.

## 4.3 Limitations of the proposed methodology

The proposed methodology for analysing high-frequency stream chemistry data is based on event-scale variations of the concentration of different solute pairs (concentration–concentration variations). We propose this methodology because it can add useful information to that gained from other commonly used methodologies (such as EMMA or $c - Q$ analysis). However, it also has its limits, especially in terms of the criteria used to define the different variation patterns ($R^2$ and its threshold of 0.8) and the difficulty in interpreting solutes with a complex and invariant variation pattern.

Another limitation of our analyses relates to the selection of analysed storm events. Due to technical challenges (pump failure, filter clogging, power outages due to lightning strikes), the analysed storm events were not fully representative of those detected based on discharge. At Kervidy-Naizin, the initial discharge of the analysed events was higher than those of the detected events based on discharge, whereas the inverse was true at Strengbach. Related to the event magnitude, the events with the smallest and largest discharge increases were underrepresented in the set of analysed events at Kervidy-Naizin, whereas at Strengbach the events with the largest discharge increases were overrepresented in the analysed dataset. Therefore, the conclusions we draw about the hydrological and (bio-)geochemical processes in the two catchments might be slightly biased.

### 4.3.1 Sensitivity of the classification

We used the coefficient of determination of the linear regression of the concentration–concentration variation of a solute pair to define a synchronous variation and, more specifically, a threshold of 0.8. Certainly, other thresholds and linearity

criteria could be used. We believe that a threshold of 0.8 is a good compromise between misidentifying complex solutes as synchronous ones and disregarding synchronous solute pairs, which are noisy due to technical uncertainties. In addition, slightly non-linear concentration–concentration variations, which could be caused by a small contribution of a third end-member, are likely classified as the synchronous variation type when using the 0.8 coefficient of determination threshold.

The choice of the 10 % variation threshold to distinguish specifically between the complex and invariant variation types has been difficult, and other thresholds could certainly be used. The challenge lies in finding a threshold that can be used for low and high solute concentrations and across different catchments and solutes while taking into consideration the solute-specific limits of quantification.

The three proposed types of solute variation (synchronous, complex, and invariant) partly overlap and are not fully mutually exclusive. In both catchments, several solutes ($K^+$ at Kervidy-Naizin; $Ca^{2+}$, $SO_4^{2-}$, $NO_3^-$, and $K^+$ at Strengbach) could not be attributed clearly to one variation type only. This could indicate that, across the different storm events, seasons, and hydrological conditions, different hydrological or (bio-)geochemical processes govern the export of the solutes and the activation of the different end-members. However, this could also be caused by the choice of the thresholds to distinguish between the different variation types, as explained above.

### 4.3.2 Difficulties in interpreting complex solute variations on the event scale

A complex solute variation pattern is not easy to interpret, as it may be caused by several different processes which cannot be distinguished using the method proposed here. Some of these explanatory processes are (1) mixing of at least three chemically different end-members, (2) intra-event variations of the end-member concentrations of a two-end-member system, and (3) non-linear reactions along the flow path between the "source" of the end-member and the stream.

The non-synchronous solutes at Kervidy-Naizin ($SO_4^{2-}$, $Ca^{2+}$, $K^+$) are the solutes with the lowest molar baseflow concentrations ($\geqq 300\,\mu mol\,L^{-1}$) of the seven solutes analysed. Therefore, spatially or temporally varying processes and reactions might increase in importance relative to simple dilution and might therefore lead to complex variation patterns. Potassium, as the solute with the lowest baseflow concentration, showed almost exclusively increasing concentrations during storm events, which were not in synchrony with any other analysed solute. These complex variation patterns of $K^+$ might be linked to its strong cycling in the vegetation and its mobilization from live or dead organic matter in the soil (Barthold et al., 2017; Knapp et al., 2020), including leaching from soils after fertilizer applications. At Strengbach, the majority of the solute pairs formed with

$Mg^{2+}$, $Na^+$, and $Cl^-$ exhibited complex variation patterns. It is difficult to interpret these results, because several factors and processes can lead to them (methodological, hydrological, and (bio-)geochemical). However, it seems that the (bio-)geochemical processes and the contributions of the various end-members are more complex at the forested catchment (Strengbach) than at the agricultural catchment (Kervidy-Naizin).

### 4.3.3 Difficulties in interpreting invariant solutes on the event scale

At Strengbach, we observed that the majority of the analysed solutes exhibited no or only limited concentration variation during storm events ($Ca^{2+}$, $Mg^{2+}$, $Na^+$, $SO_4^{2-}$). As mentioned above, this can be caused by various different processes. In addition, this limited concentration variation could mean that these solutes are not relevant for inferring event-scale processes, but it could also be related to the characteristics of the analysed events.

### 4.4 Advantage of the proposed methodology and differences from other methods

The advantage of the proposed methodology lies in its very simple application and the additional information gained when analysing high-frequency multi-elemental stream chemistry data during storm events. It can provide information about solute-specific processes in catchments without requiring a priori assumptions and enables us to identify whether two end-members are sufficient to describe the mixing pattern of a given solute pair.

Compared to the classical, forward EMMA approach that requires extensive prior knowledge to identify and chemically characterize different end-members (Durand and Juan Torres, 1996), our proposed methodology requires very little prior knowledge. It can, therefore, be applied to many catchments that are outside long-term experimental observatories. In addition, our methodology does not require the a priori assumption of conservative solutes, as is required in the EMMA approach (Christophersen et al., 1990), because even some solute reactions lead to a linear pattern on a concentration–concentration plot for a two-end-member system. However, when interpreting our results, we implicitly assume conservative behaviour of the synchronous solutes.

In comparison to the inverse EMMA based on the principal component analysis (PCA), which provides the number of potential end-members using the ensemble of all "conservative" solutes (Christophersen and Hooper, 1992; Hooper, 2003), the methodology proposed here evaluates the potential number of end-members for the solute pairs and not for the ensemble of all of the solutes together. This has two advantages. Firstly, a greater number of solutes used in the inverse EMMA also leads to a greater number of potential end-members, which is required to explain their total variation

(Barthold et al., 2011). It is therefore not straightforward to know which, and specifically how many, solutes should be used in the PCA, because both might have a large impact on the outcome of the analysis (i.e. the number of potential end-members). In the methodology proposed here, this ambiguity is reduced, because it provides the number of required end-members (two or more) for each solute pair. Secondly, based on the PCA in the inverse EMMA, it is not straightforward to know which solutes are responsible for a higher-order mixing model and which solutes require only a simpler model. Again, the methodology proposed here allows us to distinguish between the synchronous solutes that are governed by similar processes and those which are associated with more complex variation patterns that are likely due to additional solute-specific processes. Our proposed methodology can, therefore, be used to select the variables, which are then further analysed with the inverse EMMA. As synchronous solutes cannot distinguish between more than two end-members, it is likely not useful to include all synchronous solutes in an EMMA that tries to distinguish between more than two end-members.

Furthermore, the analysis of the solute ratios can provide information about the solute-specific processes at the catchment scale that are dominant or negligible. For example, the constant $Na^+/Cl^-$ ratio, observed across many different catchment compartments at Kervidy-Naizin, indicates that evapotranspiration is the dominant process that is increasing the concentrations of these solutes. In contrast, $Cl^-$ inputs from fertilizer additions or $Na^+$ inputs from weathering of primary minerals seem to be less important or negligible. Other examples are the strongly enriched $Mg^{2+}$ and $NO_3^-$ concentrations relative to $Cl^-$ and $Na^+$ at Kervidy-Naizin, which can be expected for $NO_3^-$ in an agricultural catchment but which was unexpected for $Mg^{2+}$.

Finally, our methodology could be extended in several ways or used in different applications. Instead of grouping all events together, the synchrony of the solute pairs as well as the chemistry of the pre-event end-member could be analysed as a function of the season or the hydrological antecedent condition of the catchment. Furthermore, the rising and falling discharge limbs could be analysed separately, which could be particularly interesting for the complex solutes. This could reveal whether the third end-member of the complex solutes contributes at the beginning or at the end of a storm event.

centrations, we proposed a complementary methodology that allows identification of solutes that are governed by mixing only two end-members during storm events.

The proposed methodology is based on using concentration–concentration variations during storm events (i.e. the variations of the concentrations of solutes A and B against each other) to identify synchronous solute pairs. These synchronous solutes can be interpreted as mixing of only two end-members: one concentrated/reacted end-member and one diluted/unreacted end-member. This two-end-member system can also be viewed as consisting of one pre-event and only one event-activated end-member. The concentrations of the synchronous solutes are thus spatially homogeneous in the catchment, without significant alteration in hotspots, which would otherwise lead to a second event-activated end-member and, therefore, a third end-member in total. Alternatively, the contributions from those hotspots are negligible during storm events.

Our proposed methodology can be used as a complementary tool to the inverse-type EMMA or the CHEMMA, because it indicates specifically for which solute pairs a mixture of only two end-members is sufficient and for which of them this is not the case. This information could then be used to decide which solutes to include in an inverse-type EMMA, for example, because no standard methodology for the solute selection exists to date.

We observed a synchronous concentration variation for four solutes ($Na^+$, $Cl^-$, $Mg^{2+}$, $NO_3^-$) during 56 % to 92 % of the analysed storm events at the agricultural catchment (Kervidiy-Naizin) and for two solutes ($K^+$, $Cl^-$) during 61 % of the storm events at the forested catchment (Strengbach). Therefore, these solutes are governed by only two end-members during the majority of the storm events. Even though a synchronous concentration variation could have been expected for some of these solutes, it was unexpected for the majority of them.

These results show the potential impacts of land cover, geology, topography, and climate on the relation between the stream water chemistry and the hydrological dynamics. For example, the absence of fertilizers and soils modified by agriculture at Strengbach, in contrast to Kervidy-Naizin, might be one of the reasons why no common pairs of synchronous ions were found between the two studied sites. Thus, land cover and human activities modify the hydrological dynamics as well as the geochemical signatures.

## 5 Conclusions

In this study, we presented high-frequency stream chemistry data collected with an innovative infrastructure (Riverlabs) that enabled us to sample systematically a large number of flood events, more than what is frequently collected with an automatic sampling strategy. Based on these high-frequency, multi-elemental time series of stream solute con-

**Appendix A:  Hydrographs of Kervidy-Naizin and Strengbach with the selected storm events**

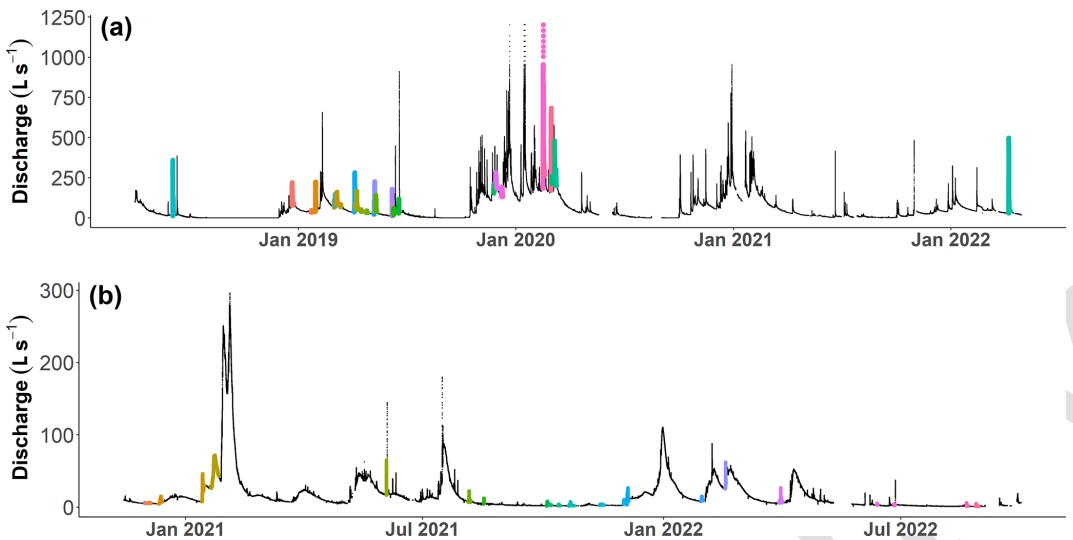

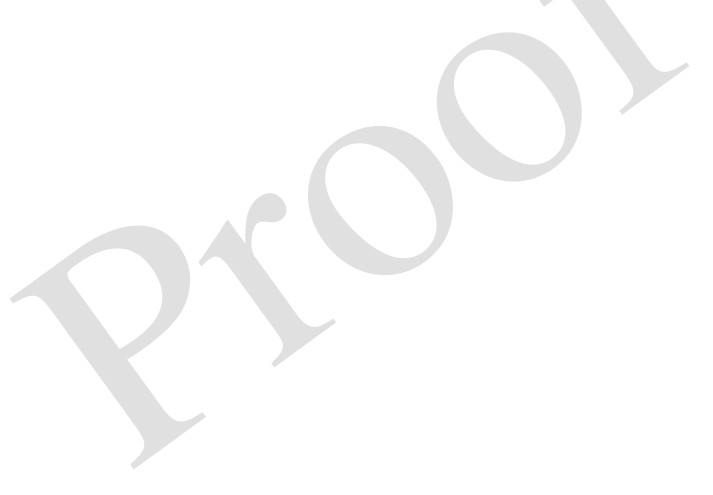

**Figure A1.** Hydrograph with selected and analysed storm events (coloured) for Kervidy-Naizin **(a)** and Strengbach **(b)**.

## Appendix B:  Examples of the chemical variation of the stream water during storm events

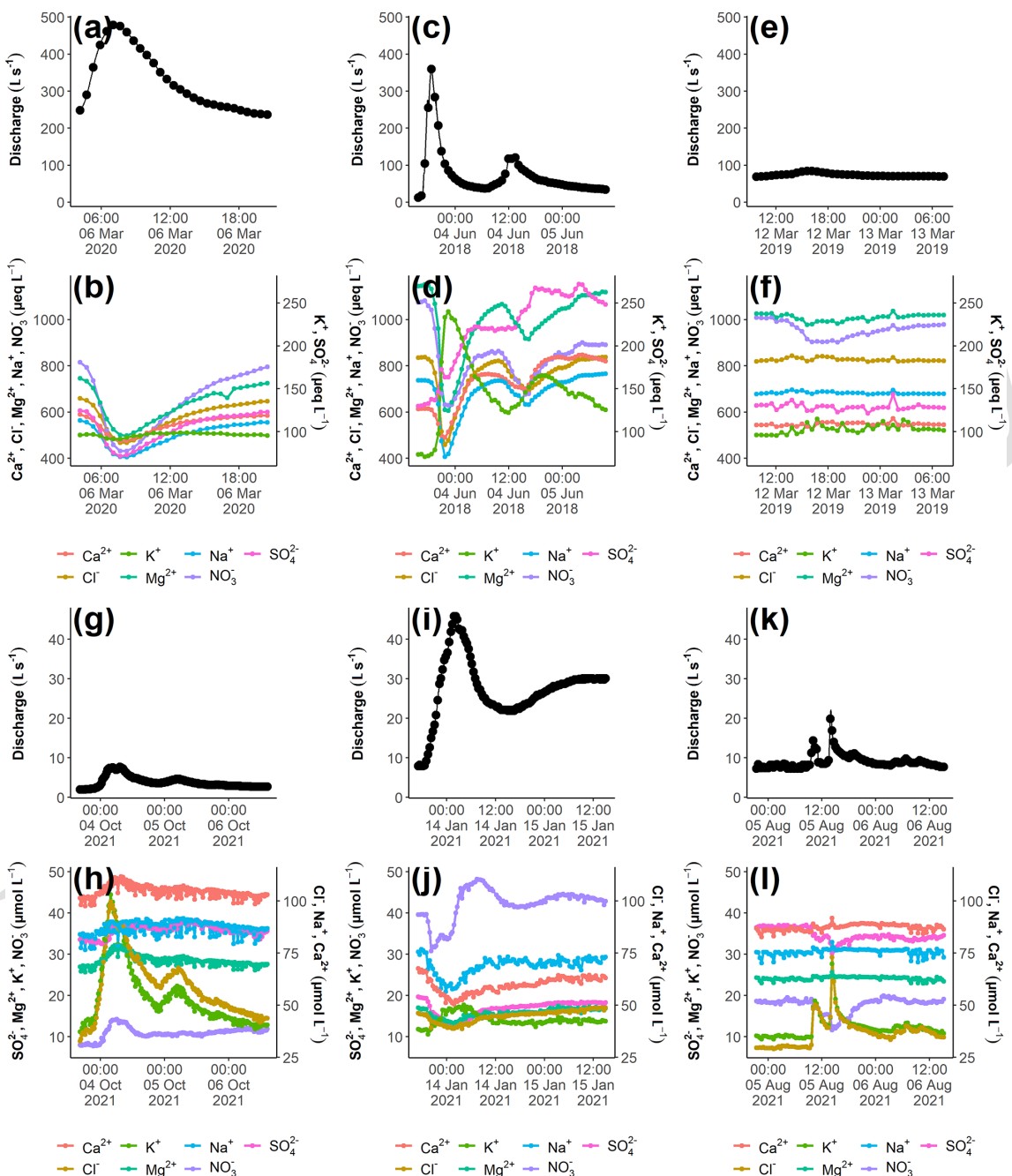

**Figure B1.** Examples of the chemical variation during three different storm events at Kervidy-Naizin **(a–f)** and Strengbach **(g–l)**. For each storm event, the discharge **(a, c, e, g, i, k)** and the corresponding chemical concentrations **(b, d, f, h, j, l)** are visualized. In order to highlight the inter-event variability, the *y* axes for the discharge and the chemical concentrations are kept constant for each catchment.

**Appendix C:  Number (and percentage) of storm events with synchronous, complex, and invariant concentration–concentration patterns**

**Table C1.** Number (and percentage) of storm events with synchronous, complex, and invariant concentration–concentration patterns for different pairs of solutes at Kervidy-Naizin. For the synchronous concentration–concentration pattern, the values are listed for two thresholds ($R^2 \geqq 0.8$ and $R^2 \geqq 0.9$). Listed are all solute pairs that exhibit a synchronous variation ($R^2 > 0.8$) in less than 30 % of the storm events.

| Ion pairs | Kervidy-Naizin ($n = 39$) | | | |
|---|---|---|---|---|
| | Synchronous | | Complex | Invariant |
| | $R^2 \geqq 0.9$ | $R^2 \geqq 0.8$ | | |
| $Ca^{2+}/Na^+$ | 6 (15 %) | 11 (28 %) | 15 (38 %) | 13 (33 %) |
| $Ca^{2+}/Mg^{2+}$ | 7 (18 %) | 11 (28 %) | 15 (38 %) | 13 (33 %) |
| $Ca^{2+}/Cl^-$ | 5 (13 %) | 10 (26 %) | 15 (38 %) | 14 (36 %) |
| $NO_3^-/Ca^{2+}$ | 4 (10 %) | 10 (26 %) | 19 (49 %) | 10 (26 %) |
| $SO_4^{2-}/Na^+$ | 7 (18 %) | 9 (23 %) | 11 (28 %) | 19 (49 %) |
| $SO_4^{2-}/Cl^-$ | 6 (15 %) | 8 (21 %) | 12 (31 %) | 19 (49 %) |
| $SO_4^{2-}/Mg^{2+}$ | 4 (10 %) | 8 (21 %) | 15 (38 %) | 16 (41 %) |
| $K^+/NO_3^-$ | 4 (10 %) | 7 (18 %) | 22 (56 %) | 10 (26 %) |
| $K^+/Mg^{2+}$ | 2 (5 %) | 6 (15 %) | 19 (49 %) | 14 (36 %) |
| $K^+/Na^+$ | 1 (3 %) | 1 (3 %) | 19 (49 %) | 19 (49 %) |
| $K^+/Cl^-$ | 1 (3 %) | 1 (3 %) | 18 (46 %) | 20 (51 %) |
| $K^+/Ca^{2+}$ | 0 (0 %) | 1 (3 %) | 11 (28 %) | 27 (69 %) |
| $K^+/SO_4^{2-}$ | 0 (0 %) | 0 (0 %) | 18 (46 %) | 21 (54 %) |

**Table C2.** Number (and percentage) of storm events with synchronous, complex, and invariant concentration–concentration patterns for different pairs of solutes at Strengbach. For the synchronous concentration–concentration pattern, the values are listed for two thresholds ($R^2 \geqq 0.8$ and $R^2 \geqq 0.9$). Listed are all solute pairs that exhibit a synchronous variation ($R^2 > 0.8$) in less than 30 % of the storm events.

| Ion pairs | Strengbach ($n = 23$) | | | |
|---|---|---|---|---|
| | Synchronous | | Complex | Invariant |
| | $R^2 \geqq 0.9$ | $R^2 \geqq 0.8$ | | |
| $Mg^{2+}/Na^+$ | 2 (9 %) | 6 (26 %) | 10 (43 %) | 7 (30 %) |
| $Ca^{2+}/Na^+$ | 2 (9 %) | 4 (17 %) | 12 (52 %) | 7 (30 %) |
| $SO_4^{2-}/Cl^-$ | 1 (4 %) | 3 (13 %) | 13 (57 %) | 7 (30 %) |
| $Mg^{2+}/Cl^-$ | 0 (0 %) | 3 (13 %) | 14 (61 %) | 6 (26 %) |
| $K^+/SO_4^{2-}$ | 1 (4 %) | 3 (13 %) | 8 (35 %) | 12 (52 %) |
| $NO_3^-/Na^+$ | 1 (4 %) | 2 (9 %) | 12 (52 %) | 9 (39 %) |
| $SO_4^{2-}/Na^+$ | 0 (0 %) | 2 (9 %) | 13 (57 %) | 8 (35 %) |
| $K^+/Na^+$ | 0 (0 %) | 2 (9 %) | 13 (57 %) | 8 (35 %) |
| $Cl^-/Na^+$ | 0 (0 %) | 2 (9 %) | 12 (52 %) | 9 (39 %) |
| $NO_3^-/Mg^{2+}$ | 0 (0 %) | 2 (9 %) | 11 (48 %) | 10 (43 %) |
| $SO_4^{2-}/Mg^{2+}$ | 0 (0 %) | 2 (9 %) | 12 (52 %) | 9 (39 %) |
| $K^+/Mg^{2+}$ | 0 (0 %) | 2 (9 %) | 12 (52 %) | 9 (39 %) |
| $NO_3^-/Ca^{2+}$ | 0 (0 %) | 2 (9 %) | 8 (35 %) | 13 (57 %) |
| $K^+/NO_3^-$ | 0 (0 %) | 2 (9 %) | 9 (39 %) | 12 (52 %) |
| $NO_3^-/Cl^-$ | 0 (0 %) | 1 (4 %) | 10 (43 %) | 12 (52 %) |
| $Ca^{2+}/Cl^-$ | 0 (0 %) | 0 (0 %) | 14 (61 %) | 9 (39 %) |
| $SO_4^{2-}/Ca^{2+}$ | 0 (0 %) | 0 (0 %) | 10 (43 %) | 13 (57 %) |
| $K^+/Ca^{2+}$ | 0 (0 %) | 0 (0 %) | 12 (52 %) | 11 (48 %) |
| $SO_4^{2-}/NO_3^-$ | 0 (0 %) | 0 (0 %) | 11 (48 %) | 12 (52 %) |

**Appendix D: Inter-event synchrony for all solute pairs**

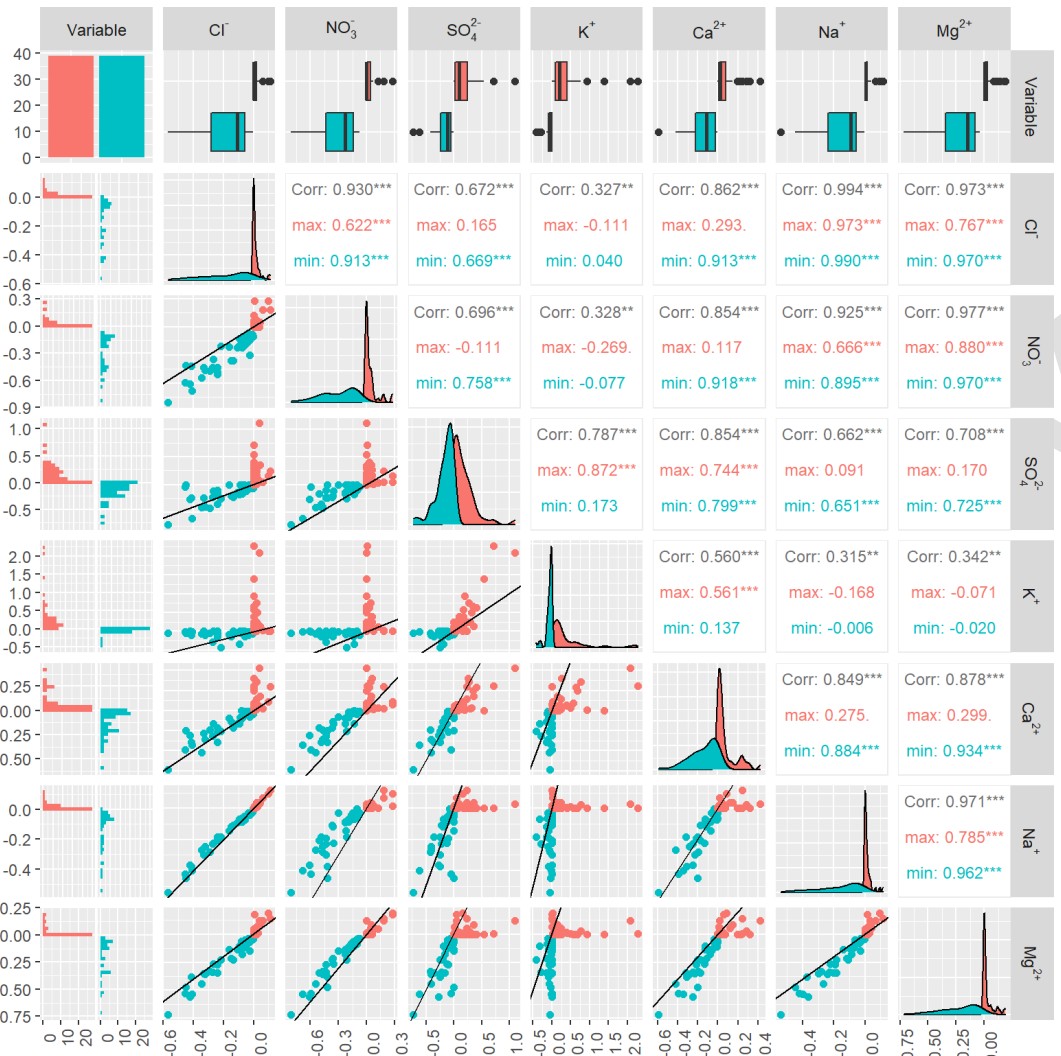

**Figure D1.** Inter-event synchrony for all 21 solute pairs for Kervidy-Naizin. Lower-left triangle of the sub-figures: relative concentration increase [red; $(C_{\text{maximum},j} - C_{\text{initial},j})/C_{\text{initial},j}$] and decrease [green blue; $(C_{\text{initial},j} - C_{\text{minimum},j})/C_{\text{initial},j}$] (%) of the maximum and minimum event concentrations in relation to the initial concentration for each event $j$. Values of 0 indicate that the maximum or minimum concentrations are equal to the initial concentration. The black line represents the 1 : 1 line. The first column, the first row, and the diagonal represent the distribution of the relative concentration increase and decrease for each solute. Upper-right triangle of the sub-figures: for each pair, the correlation coefficients are indicated for all points together (green blue plus red) in black as well as for the red and green blue points, respectively.

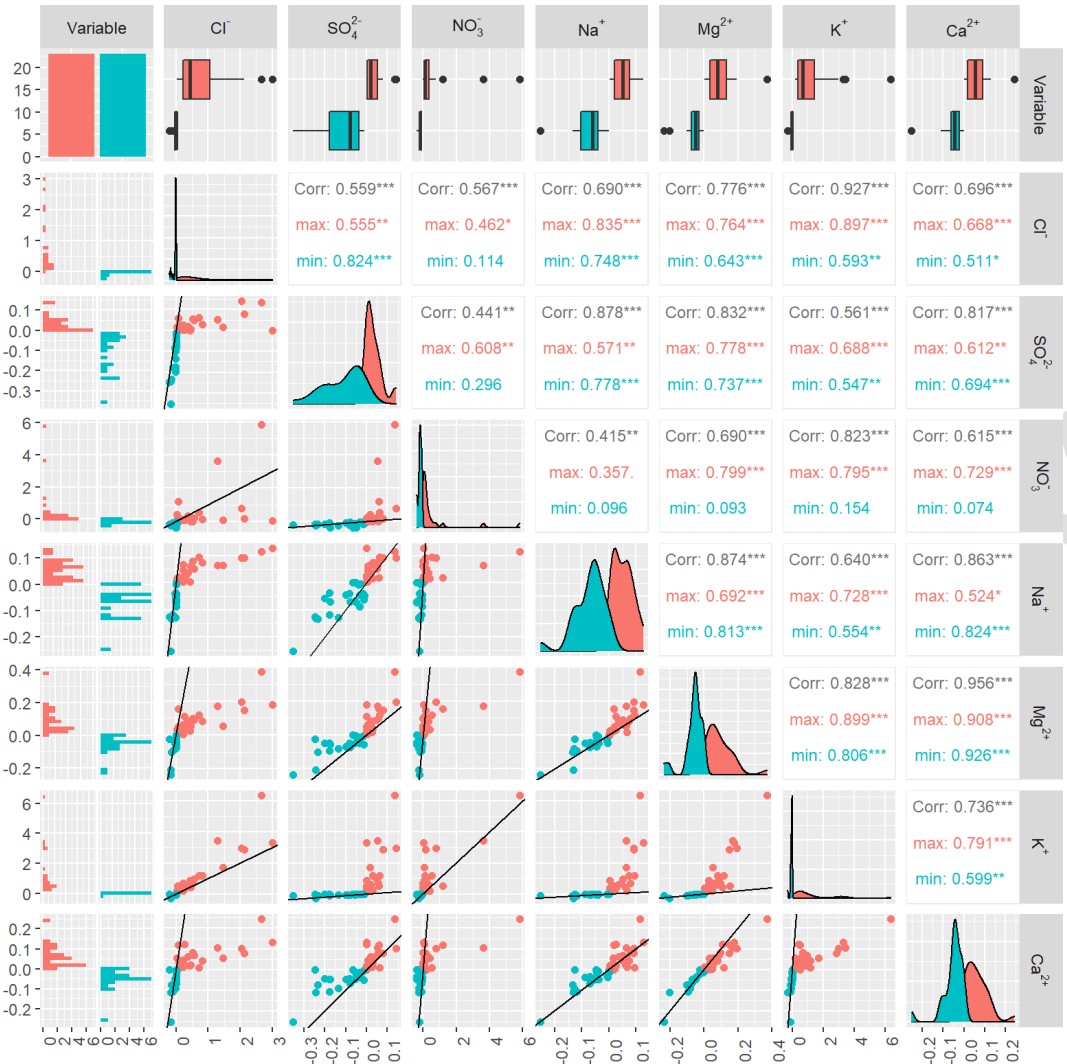

**Figure D2.** Inter-event synchrony for all 21 solute pairs for Strengbach. See the figure caption of Fig. D1 for further explanation.

*Data availability.* The data supporting the findings of this study can be downloaded from the French Research Data Repository (accessible via the following address: https://doi.org/10.57745/ENHW7M, Brekenfeld, 2024).

*Author contributions.* NB, OF, MCP, SG, HH, ACPW: conceptualization; NB: formal analysis; OF, MCP, JG: funding acquisition; NB, SC, MF, PF, CF, YH, PP, MCP, OF, APCW: methodology and investigation; NB: writing – original draft preparation; OF, SG, MCP, ACPW, CF: writing – review and editing.

*Competing interests.* The contact author has declared that none of the authors has any competing interests.

*Acknowledgements.* We thank the technical and analytical staff (Beatrice Trinkler, Laure Cordier, Sophie Gangloff, Arnaud Blanchouin, Alain Hernandez, Anthony Julien, and Pascal Friedmann) as well as Laurent Ruiz, Rémi Dupas, and Chantal Gascuel-Odoux for inspiring discussions.

*Financial support.* The Critex Programme ANR-11-EQPX-0011 funded the Riverlabs and covered most of the costs associated with their running. The people who ran the Riverlabs were staff from ORACLE, OHGE, and AgrHyS Critical Zone Observatories (i.e. staff from CNRS and INRAE). The research units UR HYCAR, IPGP, LHYGES/ITES, UMR SAS, and UMR Geosciences Rennes contributed to the running costs, especially the vehicle costs associated with regular travel to the site for maintenance, power supply, etc. The post-doctoral position of Nicolai Brekenfeld was co-funded by Region Bretagne, UMR SAS, INRAE (AQUA), and OZCAR-RI.

*Review statement.* This paper was edited by Laurent Pfister and reviewed by Karl Nicolaus Van Zweel and one anonymous referee.

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
