# Peer review of "Using high-frequency solute synchronies to determine simple twoend-member mixing in catchments during storm events"

_EGUsphere, 2023_

## Author Comment (AC1)

Referee comment #1

*We thank the referee for its thorough, detailed and constructive comments and suggestions. Below, we reply to its suggestions and questions and propose adjustments to our manuscript accordingly.*

Brekenfeld et al. propose a new method to analyze multi-elemental time series acquired at the outlet of two long-term experimental catchments using a high frequency *in situ* laboratory. This method is based on event-scale variability of concentration of solute pairs. They determine the "synchronous" behaviour of two given solutes, in that case major ions, based on their concentration relationship and state that this synchrony can define a two-end-member mixing: pre-event and event water. This methodology is divided in three steps: (1) classification based on concentration-concentration plot, (2) variation of the molar ratio at event scale and (3) the calculation of the inter-event synchrony of concentration variability. This technique is proposed as a complementary data analysis from other methodologies, such as EMMA and c-Q relationship.

This manuscript corresponds to the scope of HESS publications. However, in the form it is presented for the moment, it mainly introduces an innovative data treatment approach for interpreting solute transport at catchment scale. It does not really improve the solute transport knowledge at catchment scale for some reasons that will be discussed later on. For this reason, I suggest, instead of targeting a research article, to go for HESS technical note because this new contribution introduce a new development, which is relevant for scientific investigations within the journal scope. And this is an interesting approach, which could contribute to the revisit of the conservative tracer hypothesis. Moreover, the submitted draft still needs significant improvement before to be published.

*We agree with the reviewer, that we present an innovative data treatment methodology, but we do not only do that. We also apply it to two contrasting catchments, and interpret the results to improve our understanding of the solute transport of the catchments. In addition, we discuss catchment processes that can be deduced from the observed results. For example, we analysed the ratio of ion pairs and can deduce for the Na/Cl pair in the agricultural catchment that it is primarily influenced by precipitation and evapotranspiration and insignificantly by other catchment processes. In order to improve the clarity, that this methodology contributes to the understanding of catchment processes (hydrological, biogeochemical), we reformulate parts of the section 4.1 (discussion) and highlight the processes that advanced our understanding of the catchment processes. We therefore think that a research article is the appropriate format.*

**General Comments**

On the form, the structure and the clarity of the manuscript still needs improvements. Some parts are not located at the more judicious place (see detailed comments). The comparison between the two catchments should be more clearly separated. Indeed, the result part need a clearer structure to clarify the different solute behaviors between the two experimental catchments. Some results are provided without any introduction in the material and method part. The labelling of the figures could be harmonized and defined according to the used parameters: ions in this case.

*Ok, we try to improve the structure as detailed below. However, we do not really want to compare the different catchments, but rather, apply the method to two contrasting catchments in order to improve the knowledge of each catchment.*

*We added additional information in the material and methods section, when asked for and labelled the figures with the ions (and not the names) as proposed by the reviewer.*

All data used in a published article should be provided and/or being accessible, via online application or directly in the manuscript. This is not the case with this new contribution and I would insist to have access to all datasets used. If this information was provided and I did not find the link, I hope the authors would accept my apologies.

*We will, of course, provide the data of the retained storm events, which can be retrieved from INRAE dataverse, once we have reached the final version of the manuscript.*

One of my main concerns is about the definition of an "end-member" and the capacity we have to determine its contribution at catchment scale by only using data observed in the stream. Indeed, according to the biogeochemical complexity and the hydrological connectivity, observations made in a stream should only be extrapolated to some "near-stream" locations, like the riparian zones.

*We would like to point out, that the purpose of this manuscript and the proposed methodology is to identify the minimum number of end-members required to describe the observed solute variations for each solute pair and not to identify/characterise the end-member.*

*We agree with the referee that the stream water during storm events is primarily provided by some "near-stream" contributing areas and we therefore change our terminology from "catchment" to "contributing areas" in the manuscript, wherever we think it is suitable. However, some of the solute concentrations and their ratios might be (primarily/partly) influenced by processes acting outside of the near-stream areas, where the water infiltrates and evaporates, for example (e.g., Na/Cl). We therefore keep the terminology "catchment" when we refer to processes acting across the catchment, for example. This is one of the inherent complexities of the end-member approach, that the location of the water source (e.g., riparian zones) might not be the same as the location of the processes acting on the solutes, which are used in the EMMA.*

To my understanding, what novelty is provided in this study is a chemical identification of the common "old" versus "event" water that are used since long time in hydrological studies.

*The proposed methodology does not make prior assumptions about the origin of the end-members. It therefore cannot be used to chemically identify "old" versus "event" water. The latter one is often based on isotopic analyses and quantifies the contribution of water that fell during the analysed event (event water). Our proposed methodology, which is only based on the stream chemistry, does not allow the identification of "event" water. In contrast, it allows the quantification of the number of end-members that contribute to a storm event for a given solute pair.*

Unifying catchment biogeochemical response to hydrological dynamics based on two end-member mixing seems to be very reductive and the defined "endmembers" may not be real ones from a given contribution of a catchment compartment but more a mixing of different water (with different chemical pattern and age) along a flowline that would exist in the system during specific hydrological conditions. In other world, is it really end-member that are observed or a specific stream hydrochemistry driven by interaction with existing near-stream end-members? Even if we could assume that the event water may most of the time present similar characteristics, what about the variability of processes that could contribute to the chemistry of the pre-event water? For this reason, I suggest providing in the introduction a clear definition of what is called an end-member and present current limitation to observe end-members in a stream from the remote part of a catchment. This would allow to suggest how this new methodology could improve such limitation.

*We define an end-member as follows: "water mass (e.g., riparian zone, macro-pore solution, soil layer solution, groundwater, throughfall, etc.) with a distinct chemistry and with a distinct variation of its contribution". Furthermore, as in Hooper & Christophersen, 1990, we defined "end-members" as contributing sources that have extreme chemistry. Those end-members that can be formed by a mixture of two (or more) other end-members are not considered to be end-members. The definition of a "distinct variation of its contribution" is required, because two chemically distinct end-members, that exhibit the same variation of their contribution, appear only as one end-member with one chemical signature and not as two end-members. With "the same variation of their contribution" is meant that the contribution of one of those end-members is a constant multiple of the contribution of the other end-member ($Q_{EM1,t} = Q_{EM2,t} \times k$; $k = const.$). If for example, one end-member is always (before and during the event) contributing the double amount of water than a second end-member, then their individual contributions cannot be calculated, beause their chemistries appear as one, undifferentiable mixture in the stream.*

*To reply to the referee's question: if the "mixing of different water (with different chemical pattern and age) along a flowline" is caused by chemically distinct end-members (as defined above) and their respective, quantitative contributions are not constant multiples of each other, then these "different waters" appear as different end-members in the stream. In this case, our proposed analysis is able to differentiate between them.*

*"is it really end-member that are observed or a specific stream hydrochemistry driven by interaction with existing near-stream end-members?" That question always arises when EMMA's are conducted or during the interpretation of their results. It is difficult to answer that question. However, we think that analysing different solute pairs separately, as proposed by our methodology, might help to answer that question, because different pairs provide information about different hydrological or biogeochemical processes.*

*The novelty of this methodology is, that is does not require prior knowledge of the catchment or assumptions about the solutes and that it is very simple but very informative. It can be applied at sites, where only time series of stream water concentrations are available (does not require analyses of soil solutions or groundwater, for example).*

*We add further details in the abstract, section 4.4 and the conclusion to highlight the novelty of our approach, especially in comparison with the "forward" and "inverse" EMMA approach. In addition, we provide our definition of an end-member in the introduction. Finally, our methodology provides guidance for the pre-selection of the variables that could be useful in an inverse EMMA approach.*

The method is based on statistical threshold to justify the existence of some ion relationships and propose a paired chemical element classification in three groups: synchronous variation, invariant and complex variation. Then the observed synchrony is used to explain the processes that drive the mobility of the paired of synchronous elements to the outlet of the catchment. In my opinion, based this methodology only on statistics is a mistake because the main drivers of the hydrochemistry at event scale is the hydrological state of the catchment - meaning the status of the water storage when the rain start – and the season – meaning the activity of the vegetation and related living (micro)organisms. The statistical choice that is done here is not able to justify properly the link between observed relationship and determination of detailed and specific processes that could explain the given relationship. I think that the difficulty of this methodology relies in the selection criteria based on the number of events and not on the typology of events that would take into account hydro-climatological state of the catchment during the selected events. An event typology that relates to

event characteristics and hydrological dynamics (connectivity, storage dynamics…) in the catchment would have been more appropriate to link the c-c relationship to catchment process functioning. For instance, this would inform about the potential connectivity between functional compartments inside the catchment and the potential water mixing happening close to the stream during contrasted seasons. With the long-term hydrological timeseries that may exist at these two experimental catchments, placing the selected events in a more general hydro-climatological context would be an important added-value to this study and to the community. I wonder if the authors could go deeper in this suggestion.

*We agree with the referee that the hydrological and biogeochemical processes in catchments are very complex and that any perceptual representations of them are simplifications. We chose the coefficient of determination as our statistical metric, because it is one of the simplest measurements of the fit between a dataset and a linear regression. As outlined below, other metrics and thresholds could and can be used. Our aim was to differentiate between linear and non-linear bivariate concentration relationships. Once a linear relationship is observed, it can be concluded that a two end-member system is sufficient to describe its variation. We then underline interpret possible processes, which might be responsible for this two end-member system. The interpreted processes are not linked to the statistical method that was used to determine a linear relationship. The interpreted processes are based on the hydrological, geochemical and biogeochemical literature as well as additional observations in the catchments.*

*For simplicity, we decided to group all events together, in order to establish general, inter-seasonal and inter-annual patterns of the solute variations. Certainly, additional analyses can be conducted by separating the events based on their hydrological state, the season or others. Furthermore, the percentages given in table 1 allow the reader to recognize, that only a certain percentage of the events show a linear relationship for a given solute pair. We used the terms "synchronous" or "complex" solute pairs only as an overall classification. That does not mean that a certain solute pair, classified as a "synchronous" pair, exhibits a synchronous variation pattern during all of the analysed events. We add some words in the section 3.2 (results) to highlight this simplification of our classification.*

*We now also attributed two variables about the antecedent hydrological state (initial discharge) and the event magnitude (maximum discharge increase) to each analysed event. This allows us to compare the event characteristics of the analysed events with the characteristics of all the events detected in the same period. This provides than a measure of how representative our analysed events are.*

**Specific/detailed comments**

Introduction

Lines 69-73: this part should be included in the material and methods

*Ok. We moved lines 69-71 to the material and methods section. However, we left lines 71-73 in the introduction, because we think a short sentence about the overall aim of the study is useful at the very end of the introduction.*

Material, methods and site descriptions

Line 94: replace $Ca^{2+}$ by "Ca" or "calcium"

*Ok.*

Lines 105-110: Is alkalinity also measured in the in situ laboratory system? How did you check the ionic balance in the samples?

*No, alkalinity was not measured in the in situ laboratory and we did not check the ionic balance. However, validations and calibrations of the ion chromatography system were regularly conducted in order to validate (or not) the measured ion concentrations. In addition, grab samples, analysed on another machine, were used to cross-validate the measurement values from time to time.*

Lines 117-162: How did you define the two threshold values used to differentiate the three relation types?

*These thresholds are only used approximatively and other threshold could and can be used. As outlined in the discussion, other thresholds can be used, with a trade-off between the precision of the measurements (lower precisions requiring a lower threshold) and the ability to detect small contributions of a third end-member (requiring a higher threshold).*

*The thresholds used in this methodology therefore depend on the precision of the measurements and the willingness (or not) to detect (very) small contributions of a third end-member. We thought that an $R2$ of 0.8 might be a good compromise. In addition, we also provide the results for another threshold ($R^2>0.9$) that indicate, that the classification is not directly linked to the used thresholds.*

Lines 163-165: I would agree for each individual site but how the different delays (distance from the sampling to the in situ laboratory and membrane saturation) could affect the comparison between the two sites?

*Yes, the reviewer is right that the delays are different for the two sites. But we are not comparing directly the two sites. We apply the methodology and interpret its results separately for each site. We finally compare the results and conclusions (which are the synchronous ions?) between the both catchments but never the raw concentration variations.*

Line 173: "Parts of the catchments" should be replaced by "near stream parts/contribution areas"

*Ok. We replaced « parts of the catchments » with « contribution areas »*

Line 180: in what context were those two examples taken? Are they comparable with the context of the studied catchments?

*The study by Anderson et al., 1997, was conducted in a very steep (40°-45°), very small (860 m²), forested "unchanneled valley", with a high rainfall amount of around 2000 mm yr$^{-1}$, with highly conductive ($K_{sat}$ of $10^{-3}$ m s$^{-1}$), organic-rich soils.*

*The study by Hill 1993, uses samples from surface peat and litter (needle-leaf forest) from a groundwater connected headwater swamp, with a calcium bicarbonate rich aquifer.*

*We added a sentence, mentioning some details about the studies and highlighting that the contexts might not be similar to the context of the agricultural catchment.*

Lines 184-192: this paragraph is not clear to me because it would need more specific example of what processes are considered and what compartments/end-members are taken into account. The processes you are considering here are driven by biogeochemistry, water transit time and connectivity at the same time. What about the consideration of mixing water from different end-members in the pathway to the stream?

*The processes we are describing here (for example the effect of evapotranspiration) are not specific to some compartments or end-members, but are processes that keep ion ratios constant (or change them) in general. We try to give an overview of the general processes in catchments that could lead to constant or variable ion ratios and that are not specific to certain catchment compartments or end-members.*

*Certainly, different biogeochemical processes, water transit times and connectivity can have an impact on the water chemistry. However, not all elements might be impacted by these processes. That is where our methodology can shed light on identifying those solutes, whose concentrations change due to more complex biogeochemical processes in the catchment.*

*Mixing of different end-members on the path towards the stream can be identified with our methodology, under certain conditions: 1) If the end-member chemistries are different and not a linear combination of the "extreme" end-members and 2) if their relative contributions vary during the course of the storm events. Under these conditions, the end-members can be separate.*

*If the chemistries of the end-members, that mix on the path towards the stream, can be described as a mixture of other, more extreme end-members, than, certainly, these additional end-members will not be detected because they are "hidden" behind the mixing line of the other two, more extreme end-members.*

*Furthermore, if the two end-members that mix on the path towards the stream keep a constant relative contribution during and after the event, than, again, it is not possible to distinguish them. For example, let's say one of the end-members is always contributing 100% more water to the stream than the other end-member. If that contributing volume ratio remains constant throughout the event, the two end-members cannot be distinguished. But, under this strict condition, it is likely not even useful to try to distinguish these two end-members, because they behave like a single one.*

*We tried to make this paragraph a bit clearer by providing an example.*

Line 196: Why looking only at the denomination ion?

*We apply this method only to the synchronous ions. It therefore does not matter, if we use the denominator or the numerator to define the « peak value ». We added an additional explanatory sentence.*

Results

Lines 200-204: this part should be in the methodology in §2.4.1. It could be interesting to know the total number of events that were used for this selection to evaluate some kind of "success rate" for both in situ laboratory in the two different catchments.

*Ok. We have now calculated and included the « success rate ». However, we would like to mention that our stream-bank field laboratories were prototypes, which were only recently developed. In addition, they had to be adapted to the local conditions of the streams, which was a lengthy process. This is the reason for the "low" success rate.*

*We have now moved the information about the number of analysed events into the material and methods section.*

Lines 211-213 are redundant and could be removed.

*Ok, we removed these sentences.*

Lines 214-227: I do not really understand this analysis. Why and how the thresholds of % event is defined and how this impact the interpretation. Events that represent <5% should not be also of interest to explain different dynamics and link to processes functioning?

*For a first, overall classification we selected a threshold of 50%. Since all of the possible six solute pairs made of Na, Cl, Mg and NO3 showed a synchronous behaviour in at least 56% of the events, we classified them as "synchronous ions". Certainly, it would be very interesting to also analyse the events that did not behave "as usual", for example events that do not exhibit a synchronous variation of the synchronous solutes. Here we simply explained why solutes are classified as "synchronous" together or not. For example, K is not classified with the "synchronous solutes" as it varies in synchrony with Na, Cl, NO3 or Mg for only 5% of the events.*

*We now added an additional paragraph in section 4.4 highlighting potential extension of the analyses (such as analysing "non-usual" events).*

Line 215: from where is coming the value 56%

*The 56% is coming from Table 1, in the line of the NO3/Na pair, in the column of the R2>0.8 at Kervidy-Naizin. We included a first bracket listing the six solute pairs and a second bracket indicating that we are referring to the 0.8 threshold.*

Line 219: Cl and Mg not presented in this part and should be replaced by Ca

*For clarity, we decided to only list pairs in this table with an occurrence percentage of 30% or more. We added a table with the values of all other pairs in the appendix.*

Line 220: K not shown in the table

*See our previous comment.*

Line 256: molar ratio in the rain and the throughfall are not presented in the methodology part. This should be added. Were the rain and throughfall sampled accordingly to the stream sampling frequency? If not how can you be sure that you capture the real variability of chemistry in those input samples and that this one can be compared with the high-frequency variability you observed in the stream?

*No, the rain and throughfall were sampled much less frequent than the stream (often, monthly composite samples). In addition, this sampling scheme does not capture the high-frequency variability as it was measured in the stream. We added a sentence highlighting, that we are comparing the high-frequency variability in the stream with the long-term measurements in the rain and throughfall.*

*We added further details and references about the rain and throughfall measurements into the material and methods section. We do that by highlighting that the catchments are research observatories and reference the articles from which the data were taken and where further methodological information can be found.*

Line 264-265: this is not true for the Strengbach catchment.

*We do not fully understand, why the reviewer thinks that this is not true for Strengbach. As indicated in figure 4.b), the K/Cl ratio increases slightly (grey boxplot) from the start to the peak,*

*with a median increase of 0.05 and an IQR of 0.035 – 0.09 (grey boxplot). It is therefore true that the ratio increased during the storm events. As the median ratio in the throughfall with 1.2 (IQR : 0.5 – 4.3, written as a text in the figure 4.b.) is higher than the ratio in the stream (during the start and the peak) it is equally true, that the ratio moves towards the ratio of the throughfall.*

*However, in case the reviewer is concerned about the term « moving towards », we replaced it by « getting closer to ».*

Lines 272-278: should be part of the methodology

*We moved large sections of it into the methodology section.*

Line 284: I see in the figure C1 0.69-0.87 and 0.62-0.88 instead of the values indicated in the text.

*For each event and each ion, we calculated two values in order to better evaluate the complex ions: one for the strongest relative concentration increase (red points in Figs. C1/C2 and Fig. 5) and one for the strongest relative concentration decrease (turquoise points in Figs. C1/C2 and Fig. 5). The values in this sentence are referring to the correlation coefficients of the turquoise points (those, indicating the strongest decrease/dilution) as indicated in line 283. The turquoise correlations coefficients in Figs. C1 and C2 are referring to those points.*

*We clarified the sentence by highlighting that we are only referring to the points of the strongest dilution (turquoise).*

Discussions

Line 305: I do not understand how you can prove the spatial homogeneity of water chemical signature at catchment scale if you only have observation in the stream. What you observe actually are processes that happen in the "near-stream" environment. You only should focus on those areas that stay closer to the stream network. You cannot infer about the complex reactivity and mixing that may happen inside the catchment and which are driven by the water saturation state and by the flowpaths connectivity in the catchment subsurface.

*Ok. We added the word "contributing" into this sentence ("different contributing parts of the catchments").*

Lines 306-308: I find this process description quite weak in this paragraph and I suggest to remove this sentence because you try to relate the element ratio to the processes in the following parts.

*We decided not to remove this sentence. Instead, we added further explanation about how we used the ratios plus the information about the synchrony to draw conclusions about the importance of certain hydro-biogeochemical processes in the catchment. We hope, this gives the readers the tool to reach similar conclusions by themselves.*

Line 310: no information is provided in the methodology about the piezometer and soil solution.

*As mentioned in one of our previous replies, we now added that the catchments are research observatories and provide references to the articles, where further information can be found.*

Line 311: interpreting all the concentrated (old water) contribution with evapotranspiration needs more explanation. Is it a seasonal evolution of this relationship? Did you only observe this during the vegetative period (ok for the evapotranspiration) or did you also observe this in winter (another explanation should be found)?

*We observed for the Na/Cl pair a relatively constant ratio across many different catchment compartments (rain, soil solution, groundwater, stream water). Since neither of the two ions are recycled in the vegetation, evapotranspiration increases the ion's concentrations with a constant ratio. During different seasons, the concentrating effect of the evapotranspiration might be different, but the ratio should remain constant. That is exactly what we observe. For this reason, we think, that evapotranspiration is leading to the concentrated end-member.*

Line 321: the Strengbach should be removed from this part dedicated to agricultural catchment. This would reduce the confusion that already exists in this discussion. The structure between the element ratios and the linked processes should be improved.

*Ok. We moved this sentence further down into the discussion into the section of the forested catchment.*

*We hope that the sentences we added in order to explain the effect of the evapotranspiration (previous comment), will help to improve the link between the processes and the ratios. We, furthermore, add a sentence with further information about the processes leading to ratios of solutes that are not identical to the rain ratio (production and/or consumption processes).*

Line 340: how deep are this groundwater in the catchment?

*We added this information into the material and methods section.*

Line 342: Can the contribution of the denitrification generalized during all seasons? what would be the seasonal effect on the contribution of the denitrification? Would you expect having the same contribution from this process in summer and winter?

*The effect of denitrification on the nitrate concentration is very likely different during summer and winter. The proposed methodology would allow to further investigate different temporal scales, such as seasonal variations. However, for clarity, we decided to focus on the overall, inter-annual pattern of the solute variations in this manuscript.*

Line 356: rain data are not used in this study: might be throughfall?

*Yes, as visualized in Fig. 4, rain and throughfall data was used for comparison in this study. As mentioned above, we have provided further detail about its measurements in the material and methods section and/or referred the reader to other, relevant articles.*

Line 387: the link between hydrochemistry and hydrology in this study is missing and would have been an important new contribution and strengthen the proposed methodology.

*We now elaborated a bit more the link between the hydrology and the chemistry as well as the catchment processes we can infer from the results.*

*In fig. 4, we added the long-term concentrations of other catchment compartments, such as groundwater in the agricultural catchment and soil solution for the forested catchment. In addition, we added into the SI one bivariate concentration plot for each catchment, including the chemical signatures of some other catchment compartments.*

*The synchrony of a solute pair and the variation of the ratio during an individual and between different storm events, can provide important information about biogeochemical processes in the catchment. For example, a constant ratio during an individual and across many different storm events that is very close to the ratio of the rain, strongly indicates that no biogeochemical processes consume or produce these ions (other than those that purely dilute or concentrate).*

*That is, for example, the case for Na and Cl at Kervidy-Naizin, where our analyses established that, contrary to previous studies, the inputs of Cl by agricultural activities and of Na by weathering are negligible. Another example is the unexpected observation that there is only one event-activated end-member of NO3 and not several. Several event-activated end-member could be expected when the activation of "hot spots" and "cold spots" for denitrification are assumed.*

*In section 4.4, we now added further information about the potential hydrological and biogeochemical interpretations that can be drawn when using our methodology.*

Lines 395-396: not clear

*We tried to improve the clarity of this sentence by restructuring it and by adding brackets about the threshold values.*

Lines 401-409: the technical uncertainty should be similar for all samples or you should explain how this could affect your methodology. Is it not more the choice of the statistical criteria that would impact the sensitivity of your classification?

*Even though the coefficient of determination might has its limitation, we are not aware of a statistical criterion that would be similarly simple and easy to understand. However, we are open to suggestions from the referee.*

*The fact that we do not have a representative set of storm events for each season has likely a much stronger impact on our classification than the statistical criterion, we chose.*

Lines 416-417: you are providing references from a forested catchment to explain processes in the agricultural one, is it really relevant?

*We have now deleted that part of the sentence, which is referring to the specific influence of biological processes on the concentrations of SO4, Ca and K.*

Line 424: any reference to strengthen this?

*We added the reference of Ackerer et al., 2020, who showed a relatively strong inter-annual variation of the nitrate concentration in the springs as well as Ladouche et al., 2001, who also observed a complex variation pattern of NO3 in the stream during a storm event (diluted in the first and concentrated in the second part of the event). We also add that the transformation by the vegetation and the microbial activity might have an important impact on the inter-annual variation of the NO3 concentration.*

Lines 433-440: this should be developed and this is also why an event typology should have been used to more efficiently understand the dynamics and related processes observed with this new methodology.

*As outlined above, we have now calculated the success rate of our stream laboratory by dividing the number of analysed storm events by the total number of events detected in the same period. Furthermore, we attributed to each detected storm event characteristics about the antecedent catchment condition (initial discharge) and the storm magnitude (maximum discharge increase). These attributions are then used to evaluate how representative our analysed storm events.*

*We now added these information to this section to make the point clearer that we are especially missing out on the large storm events.*

*Concerning the referee's proposition of an event typology, we think that this could be an interesting additional approach. However, we think adding an event typology might draw the attention away from the very simple but powerful methodology we are presenting.*

Lines 442-458: You are proposing a very pertinent and novel approach but according to all the questions I highlighted your method present similar limitation than EMMA and PCA. For instance, you are not able to explain a detailed temporal contribution of the real different end-member at catchment scale and you mainly provide more precision to link the chemical composition of the "old" and "event" water to some related processes during flood events in the "near-stream" parts of the catchment.

*We are glad that the reviewer finds our approach pertinent and novel, but we would like to highlight that there are key differences between our methodology on the one hand and EMMA ("inverse" or "forward" by using PCA) and other more sophisticated techniques such as CHEMMA (convex hull EMMA) on the other hand.*

*A classical "forward" analysis requires that the end-members are known and chemically characterised before the EMMA can be conducted. That means the catchment is forced into a conceptual model with 2, 3, or more end-members. This is not the case with our methodology. It does not require the prior determination and characterisation of the different end-members. In addition, our methodology does not assume a certain conceptual model with a given number of end-members. It is only based on the observed chemical variations.*

*In the "inverse" analysis, the dimensionality of a dataset is calculated, which is then used to determine the minimum number of end-members required. However, the selection of the variables used in the analysis is not straightforward and a common selection procedure does not exist. That is where our methodology proposes a first solution. It analyses all variables and provides for each of them a first indication what each variable could add to the analyses.*

*We add further details in the abstract, section 4.4 and the conclusion to highlight the novelty of our approach, especially in comparison with the "forward" and "inverse" EMMA approach.*

Tables

Table 1: You should stay consistent and or showing all the pairs you studied or only the ones you discuss (why keeping NO3/Na?). Presenting all analyzed ratios would allow having a larger overview about expected variability from the full dataset.

*As mentioned above, for the reason of clarity, we only include solute pairs in the table that exhibited a percentage of synchronous behaviour of at least 30%. We added another table with all the remaining pairs in the SI.*

 Figures

Figure 2: the x and y axis should be replaced by the ions instead of having the full name of the element.

*We replaced the names by the ions.*

Figure 3: same remark as Figure 2

*We replaced the names by the ions.*

Figure 4: Why Mg/Ca ratio is not presented? The "pairwise difference" is not presented in the legend

*We added the "pairwise difference" into the legend. Mg/Ca was not presented, because we did not classify it into the synchronous solutes.*

Figure 5: same remark as figure 2. The synchrony is not clear in Fig. 5b. Should it not be 39 and 23 events (then dots) for the 2 catchments in both graphs?

*We replaced the names by the ions.*

*In addition, we added further information in the figure caption. Indeed NO3 and Ca at Kervidy-Naizin (Fig. 5.b) are only synchronous for the "diluting" part. However, as Ca is a complex ion, parts of the variation (red points) are not synchronous with NO3.*

*There are 39 and 23 events, but two dots (one each color) for each events. For each event and each ion, the red dot is indicating the strongest relative concentration increase and the turquois dot is indicating the strongest relative concentration decrease.*

---

## Author Comment (AC2)

Referee comment #2

*We thank the second referee for his constructive, useful and detailed comments and suggestions to our manuscript. Below, we reply to his questions and suggestions in detail.*

**Critical review of the paper's discussion on solute synchronies and end-member mixing**

**Introduction**: The paper aims to determine the minimum number of end-members required to explain the variation of stream water solute concentrations during storm events based on the synchronous or asynchronous behaviour of different solute pairs. The authors propose a novel methodology that uses high-frequency solute synchronies to identify simple two-end-member mixing scenarios and more complex higher-order mixing scenarios. They apply this methodology to two French catchments with contrasting characteristics and analyse several major ion pairs on the event scale.

**Event-scale concentration-concentration pattern**: The authors present the results of their methodology for each catchment and each solute pair, using concentration-concentration plots and histograms of the slope and intercept of the linear regression between the solute concentrations. They classify the events into three categories: (1) events that can be explained by a simple two-end-member mixing model; (2) events that require a higher-order end-member mixing model; and (3) events that show no clear pattern or relationship between the solute concentrations. They discuss the possible causes and implications of these categories, such as the influence of precipitation amount and intensity, the spatial variability of solute sources and flow paths, the occurrence of biogeochemical processes, and the uncertainty of the end-member composition.

**Strengths and weaknesses**: The paper's discussion on the solute synchronies and end-member mixing is comprehensive and informative, as it provides a detailed description and interpretation of the results for each catchment and each solute pair. The authors also acknowledge the limitations and uncertainties of their methodology and data and suggest ways to improve them in future studies. However, the paper's discussion could be improved by comparing and contrasting the results with results obtained using EMMA.

*We are pleased to hear that the reviewer finds our discussion comprehensive and informative. Concerning his last suggestion, we would like to highlight, that the objective of this manuscript and the presented methodology is to determine the minimum number of end-members that is required to explain the variation of a certain solute pair and not to identify the end-members and their chemical signature. The methodology we propose here can provide information to conduct an EMMA, particularly an inverse EMMA.*

*We have now tried to clarify this complementarity and possible link with the inverse EMMA in section 4.4.*

**Conclusion**: The paper's discussion on solute synchronies and end-member mixing is a valuable contribution to the field of catchment hydrochemistry, as it introduces a new methodology that can help identify the minimum number of end-members and the hydro-biogeochemical processes that affect the stream water solute concentrations during storm events. However, the paper could benefit from a more extensive comparison with other studies that have addressed similar research questions in order to provide a broader perspective and context for the results and implications. A good example of current views on this topic is CHEMMA (Convex-Hull End-Member Mixing Analysis).

*We now mentioned the use of CHEMMA in the introduction (Fei & Harman, 2022, HESS) and elaborate a bit further, what the "forward" and "inverse" EMMA and the CHEMMA can and cannot do. This leads to the introduction of our proposed methodology.*

*In the discussion, we re-take the main differences of the EMMA/CHEMMA on the one hand and our proposed methodology on the other hand and highlight what our methodology can add.*

Minor revisions:

Given that both PCA in EMMA and the proposed methodology operate under the same assumptions of conservation of mass and non-reactivity of solutes and both interpret variance in solute concentrations as evidence of hydrodynamic mixing, could the authors elaborate on the unique contributions of their proposed methodology? Specifically, while PCA in EMMA not only identifies end members but also provides information about the main solutes contributing to each end member through the loadings of the principal components, it is not immediately clear what additional insights the proposed methodology offers. Could the authors provide further justification for the introduction of this new method?

*The uniqueness of our proposed methodology is, that it does not require any prior assumptions, but that it is purely based on simple observations. In fact, the methodology does not necessarily require assumptions about conservation of mass or non-reactivity. To provide an example, first-order reactions also lead to a linear relationship on a concentration-concentration plot with two end-members. In addition, the proposed methodology does not require any prior knowledge about the catchment or additional measurements.*

*The PCA analysis, in contrast, requires a pre-selection of the variables, which are used in the PCA (in terms of the number and identity of the variables used). Choosing the right number of (conservative) variables to be used in the PCA can be challenging and can have an impact on the outcome (Barthold et al., 2011, Water Resours. Res.)*

*We have now strengthened this point in the introduction and the discussion.*

In the figure 2 caption, it should just be mentioned that the colours of the data points correspond to different consecutive flood events.

*We have now explained in the figure 2 caption, that the coloured data points are referring to individual measurement points.*

The intext reference in line 200 showed an error.

*Ok. Corrected.*

I would suggest performing a PCA on the data in order to see if these interpretations discussed about solute behaviour makes sense in terms of the covariance of parameters.

*We could perform a PCA with the four synchronous solutes at Kervidy-Naizin (Cl, Na, Mg, NO3), for example, to see if two end-members are sufficient even if all four solutes are taken together. However, we do not think that this would improve the understanding of our proposed methodology, but would rather make the manuscript more complex.*

*Instead, we now included in a paragraph about further extensions of this methodology (section 4.4.), that a PCA could be conducted on the synchronous ions to verify, if all synchronous solutes together also only require two end-members.*

Suggestions

This technique is only relevant in specific cases of streamflow generation since it is based on the premise that there are only two end members, which is only true when the water sources are near the stream.

*As mentioned above, the presented methodology is not based on any premises and does not require any prior assumptions, which is the main advantage of this methodology. As such, it can be applied to any case. Once the methodology is applied to a certain dataset, conclusions can be drawn about the minimum number of end-members required for a certain solute pair. A two end-member system, therefore, is not a requirement but the conclusion of the applied methodology.*

*We have now highlighted this point in the discussion, section 4.4.*

The technique does not account for variance in the pre-event end member, which will most likely change as the system wets up and flowlines extend further away from the stream.

*We agree with the referee, that in the currently presented form, the proposed methodology does not address the inter-event variance of the pre-event end-member explicitly, which could be easily added. However, fig. 4 hints at this variance, by showing the inter-event variances of the initial and peak molar ratios (red and green boxplots). In addition, this figure indicates whether the initial and peak molar ratios differ between each other despite their inter-event variance (grey boxplot).*

*However, analysing the variance of the pre-event end-member, as a function of time, season, hydrological conditions etc. are viable extensions of the proposed methodology. We have now added this potential extension in section 4.4.*

Sensitivity of the classification:

**Choice of Threshold**: To address the arbitrariness of the threshold, the authors could conduct a sensitivity analysis. This would involve varying the threshold and observing how the classification results change. This could provide a more robust justification for the chosen threshold or suggest a different optimal value.

*This is a good idea, which we covered to some degree, by presenting in table 1 the ranking for two different thresholds ($R^2 > 0.8$ or $R^2 > 0.9$). The table indicates that changing the threshold slightly would not change the classification of the variables.*

*This threshold is only used approximatively. As outlined in the discussion, other thresholds can be used, with a trade-off between the precision of the measurements (lower precisions requiring a lower threshold) and the ability to detect small contributions of a third end-member (requiring a higher threshold).*

*The thresholds used in this methodology therefore depend on the precision of the measurements and the willingness (or not) to detect (very) small contributions of a third end-member. We therefore think that the determination of the threshold values is best done manually by visually inspecting the measurement noise, for example.*

**Non-linearity**: To account for non-linearity, the authors could consider using non-linear regression models or machine learning techniques that can capture complex relationships in the data. This would allow them to classify solute variations without assuming linearity.

*The linearity is a central part of this methodology, because it is a consequence of a two end-member system. The methodology does not assume linearity. It allows to observe linearity and to draw conclusions about the number of required end-members.*

*It certainly would be possible to characterize the non-linear relationships in more detail. However, we do not think that this would add further information about the end-members or the catchment processes. We, therefore, do not address this point in the manuscript.*

**Overlap of Classification Types**: To address the overlap of classification types, the authors could consider using a probabilistic classification scheme. Instead of assigning each solute to a single category, they could assign probabilities to each category, reflecting the degree of certainty in the classification. This would acknowledge the complexity of the system and the potential for solutes to exhibit characteristics of multiple categories.

*This is an interesting idea. We would like to highlight, though, that the "invariant" category is based on individual solutes, whereas the "synchronous" and "complex" variation categories are based on pairs of solutes. Strictly speaking, we, therefore, cannot create a probabilistic classification scheme for each solute, but rather for solute pairs.*

*We now included in the SI a table (all 7 solutes in 7 rows and 7 columns), indicating for each pair the percentage of synchronous, complex and invariant relationships.*

**Case of $Ca^{2+}$/$Mg^{2+}$**: For cases like $Ca^{2+}$ and $Mg^{2+}$, where there is evidence of synchronous variation but the relative variation is low, the authors could consider creating a separate category or sub-category. This would allow them to acknowledge the synchronous variation without contradicting their classification criteria.

*This is a good idea. However, instead of creating sub-categories, we now consider ranking the relationships. Firstly, invariant solutes are removed from further analyses, because they do not provide additional information. Secondly, the remaining solutes are divided into synchronous or complex relationships based on solute pairs.*

*We now mention that point in section 4.3.1.*

Meybeck and Moatar (2012) proposed a method for segmenting c Q curves based on the stream's median flow (q50), resulting in nine distinct c Q modalities. This method can be used to subset the chemistry data to find solute pairs that exhibit this synchronous behaviour. I am primarily interested in how the linear regression line was fitted to the data. There seem to be inflection points in the data suggesting a switching of the dominance of one end member over another. I believe fitting only one regression line may not be the best way to go about it.

*This is an interesting point, which we investigated as well. One possibility, for example, is to separate the chemical variation during the rising discharge limb from the variation during the falling limb and evaluate the linearity for each part separately. We decided not to include this separation due to the clarity of the manuscript and due to the fact that it is not possible to synchronize the chemistry and discharge time series.*

*However, we added a paragraph in the discussion about potential extensions of the methodology (separating into rising and falling limb; varying the thresholds used; variability of the pre-event end-member etc.).*

*To answer the question of the referee, for each storm event and solute pair, we used the function lm() in R studio to calculate and summary() to extract the coefficient of determination. We added this information in the material and methods section, section 2.4.1.*

"In addition, our methodology does not require the a priori assumption of conservative solutes, as it is required in the EMMA approach (Christophersen et al., 1990)." I do not completely agree with this statement. The interpretation of two end-members by looking at the co-variance of solutes very much relies on the fact that no chemical reaction takes place.

*We try to explain this point in more detail: The methodology evaluates the existence of a linear relationship on a concentration-concentration plot. If a linear relationship exists, it can be concluded that only a two end-member system is required. This is independent of whether chemical reactions take place or not, because a linear relationship of a two end-member system is observed, even if first-order reactions take place ($C_{ion\ i,stream} = C_{ion\ i,end-member} \times k; k = reaction\ constant$). Therefore, this methodology does not make prior assumptions about the conservativeness of the solutes. However, higher order reactions in a two end-member system lead to non-linear bivariate concentration relationships. The referee is right, though, that we interpret the results of the synchronous solutes as if they were conservative. We therefore extent the sentence by adding that we implicitly assume a conservative behaviour of the synchronous solutes for the interpretation of the results.*

Adding c Q graphs of the solutes discussed will help to link this work to current work revisiting this concept. It will also give the reader a better conceptual feel of what is going on (flushing or chemostatic behaviour, for instance).

*Due to the variable and unknown transfer time of the water to the analytical instrument, it is, unfortunately, not possible to synchronize the discharge with the ion concentration data. It is therefore not possible to plot accurate c-Q plots.*

It would be interesting to see pH also added to the time series data.

*Similar to our previous reply, the pH and ion concentration data cannot be synchronized easily and accurately. It is therefore difficult to add the pH data to the concentration time series.*